# STOCHASTIC GRADIENT DESCENT LEARNS STATE EQUATIONS WITH NONLINEAR ACTIVATIONS

## ABSTRACT

We study discrete time dynamical systems governed by the state equation $\boldsymbol{h}_{t+1} = \phi(\boldsymbol{A}\boldsymbol{h}_t + \boldsymbol{B}\boldsymbol{u}_t)$. Here $\boldsymbol{A}, \boldsymbol{B}$ are weight matrices, $\phi$ is an activation function, and $\boldsymbol{u}_t$ is the input data. This relation is the backbone of recurrent neural networks (e.g. LSTMs) which have broad applications in sequential learning tasks. We utilize stochastic gradient descent to learn the weight matrices from a finite input/state trajectory $\{\boldsymbol{u}_t, \boldsymbol{h}_t\}_{t=0}^N$. We prove that SGD estimate linearly converges to the ground truth weights while using near-optimal sample size. Our results apply to increasing activations whose derivatives are bounded away from zero. The analysis is based on i) a novel SGD convergence result with nonlinear activations and ii) careful statistical characterization of the state vector. Numerical experiments verify the fast convergence of SGD on ReLU and leaky ReLU in consistence with our theory.

## 1 INTRODUCTION

A wide range of problems involve sequential data with a natural temporal ordering. Examples include natural language processing, time series prediction, system identification, and control design, among others. State-of-the-art algorithms for sequential problems often stem from dynamical systems theory and are tailored to learn from temporally dependent data. Linear models and algorithms; such as Kalman filter, PID controller, and linear dynamical systems, have a long history and are utilized in control theory since 1960's with great success (Brown et al. (1992); Ho & Kalman (1966); Åström & Hägglund (1995)). More recently, nonlinear models such as recurrent neural networks (RNN) found applications in complex tasks such as machine translation and speech recognition (Bahdanau et al. (2014); Graves et al. (2013); Hochreiter & Schmidhuber (1997)). Unlike feedforward neural networks, RNNs are dynamical systems that use their internal state to process inputs. The goal of this work is to shed light on the inner workings of RNNs from a theoretical point of view. In particular, we focus on the RNN state equation which is characterized by a nonlinear activation function $\phi$, state weight matrix $\boldsymbol{A}$, and input weight matrix $\boldsymbol{B}$ as follows

$$\boldsymbol{h}_{t+1} = \phi(\boldsymbol{A}\boldsymbol{h}_t + \boldsymbol{B}\boldsymbol{u}_t), \tag{1.1}$$

Here $\boldsymbol{h}_t$ is the state vector and $\boldsymbol{u}_t$ is the input data at timestamp $t$. This equation is the source of dynamic behavior of RNNs and distinguishes RNN from feedforward networks. The weight matrices $\boldsymbol{A}$ and $\boldsymbol{B}$ govern the dynamics of the state equation and are inferred from data. We will explore the statistical and computational efficiency of stochastic gradient descent (SGD) for learning these weight matrices.

**Contributions:** Suppose we are given a finite trajectory of input/state pairs $(\boldsymbol{u}_t, \boldsymbol{h}_t)_{t=0}^N$ generated from the state equation (1.1). We consider a least-squares regression obtained from $N$ equations; with inputs $(\boldsymbol{u}_t, \boldsymbol{h}_t)_{t=1}^N$ and outputs $(\boldsymbol{h}_{t+1})_{t=1}^N$. For a class of activation functions including leaky ReLU and for stable systems[1], we show that SGD *linearly converges* to the ground truth weight matrices while requiring near-optimal trajectory length $N$. In particular, the required sample size is $\mathcal{O}(n + p)$ where $n$ and $p$ are the dimensions of the state and input vectors respectively. The results are extended to unstable systems when the samples are collected from multiple independent RNN trajectories rather than a single trajectory. Our theory applies to increasing activation functions

---

[1] Throughout this work, a system is called stable if the spectral norm of the state matrix $\boldsymbol{A}$ is less than 1.

whose derivatives are bounded away from zero, which includes leaky ReLU, and Gaussian input data. Numerical experiments on ReLU and leaky ReLU corroborate our theory and demonstrate that SGD converges faster as the activation slope increases. To obtain our results, we i) characterize the statistical properties of the state vector (e.g. well-conditioned covariance) and ii) derive a novel SGD convergence result with nonlinear activations; which may be of independent interest. As a whole, this paper provides a step towards foundational understanding of RNN training via SGD.

## 1.1 RELATED WORK

Our work is related to the recent optimization and statistics literature on linear dynamical systems (LDS) and neural networks.

**Linear dynamical systems:** The state-equation (1.1) reduces to a LDS when $\phi$ is the linear activation ($\phi(x) = x$). Identifying the weight matrices is a core problem in linear system identification and is related to the optimal control problem (e.g. linear quadratic regulator) with unknown system dynamics. While these problems are studied since 1950's (Ljung (1998; 1987); Åström & Eykhoff (1971)), our work is closer to the recent literature that provides data dependent bounds and characterize the non-asymptotic learning performance. Recht and coauthors have a series of papers exploring optimal control problem (Simchowitz et al. (2018); Tu et al. (2018; 2017)). In particular, Hardt et al. (2016) shows gradient descent learns single-input-single-output (SISO) LDS with polynomial guarantees. Oymak & Ozay (2018) and Faradonbeh et al. (2018) provide sample complexity bounds for learning LDS. Sanandaji et al. (2011b;a); Pereira et al. (2010) study the identification of sparse systems.

**Neural networks:** There is a growing literature on the theoretical aspects of deep learning and provable algorithms for training neural networks. Most of the existing results are concerned with feedforward networks. Ge et al. (2017); Li & Yuan (2017); Mei et al. (2018b); Soltanolkotabi (2017); Janzamin et al. (2015); Zhong et al. (2017b) consider learning fully-connected shallow networks with gradient descent. Mei et al. (2018a); Soltanolkotabi et al. (2017); Foster et al. (2018) analyze empirical landscape of related nonlinear learning problems. Brutzkus & Globerson (2017); Zhong et al. (2017a); Du et al. (2017); Goel et al. (2018) address convolutional neural networks; which is an efficient weight-sharing architecture. Brutzkus et al. (2017); Wang et al. (2018) studies over-parameterized networks when data is linearly separable. Janzamin et al. (2015); Oymak & Soltanolkotabi (2018) utilize tensor decomposition techniques for learning feedforward neural nets. For recurrent networks, Sedghi & Anandkumar (2016) proposed tensor algorithms with polynomial guarantees and Khrulkov et al. (2017) studied their expressive power. More recently, Miller & Hardt (2018) showed that stable RNNs can be approximated by feed-forward networks.

## 2 PROBLEM SETUP

We first introduce the notation. $\|\cdot\|$ returns the spectral norm of a matrix and $s_{\min}(\cdot)$ returns the minimum singular value. The activation $\phi : \mathbb{R} \to \mathbb{R}$ applies entry-wise if its input is a vector. Throughout, $\phi$ is assumed to be a 1-Lipschitz function. With proper scaling of its parameters, the system (1.1) with a Lipschitz activation can be transformed into a system with 1-Lipschitz activation. The functions $\boldsymbol{\Sigma}[\cdot]$ and $\mathbf{var}[\cdot]$ return the covariance of a random vector and variance of a random variable respectively. $\boldsymbol{I}_n$ is the identity matrix of size $n \times n$. Normal distribution with mean $\boldsymbol{\mu}$ and covariance $\boldsymbol{\Sigma}$ is denoted by $\mathcal{N}(\boldsymbol{\mu}, \boldsymbol{\Sigma})$. Throughout, $c, C, c_0, c_1, \ldots$ denote positive absolute constants.

**Setup:** We consider the dynamical system parametrized by an activation function $\phi(\cdot)$ and weight matrices $\boldsymbol{A} \in \mathbb{R}^{n \times n}, \boldsymbol{B} \in \mathbb{R}^{n \times p}$ as described in (1.1). Here, $\boldsymbol{h}_t$ is the $n$ dimensional state-vector and $\boldsymbol{u}_t$ is the $p$ dimensional input to the system at time $t$. As mentioned previously, (1.1) corresponds to the state equation of a recurrent neural network. For most RNNs of interest, the state $\boldsymbol{h}_t$ is hidden and we only get to interact with $\boldsymbol{h}_t$ via an additional output equation. For Elman networks Elman (1990), this equation is characterized by some output activation $\phi_y$ and output weights $\boldsymbol{C}, \boldsymbol{D}$ as follows

$$\boldsymbol{y}_t = \phi_y(\boldsymbol{C}\boldsymbol{h}_t + \boldsymbol{D}\boldsymbol{u}_t). \tag{2.1}$$

In this work, our attention is restricted to the state equation (1.1); which corresponds to setting $\boldsymbol{y}_t = \boldsymbol{h}_{t+1}$ in the output equation. To analyze (1.1) in a non-asymptotic data-dependent setup, we assume a finite input/state trajectory of $\{\boldsymbol{u}_t, \boldsymbol{h}_t\}_{t=0}^N$ generated by some ground truth weight matrices

---

**Algorithm 1** Learning state equations with nonlinear activations

---

1: **Inputs:** $(\boldsymbol{y}_t, \boldsymbol{h}_t, \boldsymbol{u}_t)_{t=1}^N$ sampled from a trajectory. Scaling $\mu$, learning rate $\eta$. Initialization $\boldsymbol{A}_0, \boldsymbol{B}_0$.
2: **Outputs:** Estimates $\hat{\boldsymbol{A}}, \hat{\boldsymbol{B}}$ of the weight matrices $\boldsymbol{A}, \boldsymbol{B}$.
3: $\boldsymbol{x}_t \leftarrow [\mu \boldsymbol{h}_t^T \ \boldsymbol{u}_t^T]^T$ for $1 \le t \le N$.
4: $\boldsymbol{\Theta}_0 \leftarrow [\mu^{-1} \boldsymbol{A}_0 \ \boldsymbol{B}_0]$
5: **for** $\tau$ from 1 to END **do**
6:     Pick $\gamma_\tau$ from $\{1, 2, \ldots, N\}$ uniformly at random.
7:     $\boldsymbol{\Theta}_\tau \leftarrow \boldsymbol{\Theta}_{\tau-1} - \eta \nabla \mathcal{L}_{\gamma_\tau}(\boldsymbol{\Theta}_{\tau-1})$
8: **end for**
9: **return** $[\hat{\boldsymbol{A}} \ \hat{\boldsymbol{B}}] \leftarrow \boldsymbol{\Theta}_{\text{END}} \begin{bmatrix} \mu \boldsymbol{I}_n & 0 \\ 0 & \boldsymbol{I}_p \end{bmatrix}$.

---

$(\boldsymbol{A}, \boldsymbol{B})$. Our goal is learning the unknown weights $\boldsymbol{A}$ and $\boldsymbol{B}$ in a data and computationally efficient way. In essence, we will show that, if the trajectory length satisfies $N \gtrsim n + p$, SGD can quickly and provably accomplish this goal using a constant step size.

**Appoach:** Our approach is described in Algorithm 1. It takes two hyperparameters; the scaling factor $\mu$ and learning rate $\eta$. Using the RNN trajectory, we construct $N$ triples of the form $\{\boldsymbol{u}_t, \boldsymbol{h}_t, \boldsymbol{h}_{t+1}\}_{t=1}^N$. We formulate a regression problem by defining the output vector $\boldsymbol{y}_t$, input vector $\boldsymbol{x}_t$, and the target parameter $\boldsymbol{C}$ as follows

$$\boldsymbol{y}_t = \boldsymbol{h}_{t+1} \quad , \quad \boldsymbol{x}_t = \begin{bmatrix} \mu \boldsymbol{h}_t \\ \boldsymbol{u}_t \end{bmatrix} \in \mathbb{R}^{n+p} \quad , \quad \boldsymbol{C} = [\mu^{-1} \boldsymbol{A} \ \boldsymbol{B}] \in \mathbb{R}^{n \times (n+p)}. \tag{2.2}$$

With this reparameterization, we find the input/output identity $\boldsymbol{y}_t = \phi(\boldsymbol{C}\boldsymbol{x}_t)$. We will consider the least-squares regression given by

$$\mathcal{L}(\boldsymbol{\Theta}) = \frac{1}{N} \sum_{t=1}^N \mathcal{L}_t(\boldsymbol{\Theta}) \quad \text{where} \quad \mathcal{L}_t(\boldsymbol{\Theta}) = \frac{1}{2} \|\boldsymbol{y}_t - \phi(\boldsymbol{\Theta}\boldsymbol{x}_t)\|_{\ell_2}^2. \tag{2.3}$$

For learning the ground truth parameter $\boldsymbol{C}$, we utilize SGD on the loss function (2.3) with a constant learning rate $\eta$. Starting from an initial point $\boldsymbol{\Theta}_0$, after END SGD iterations, Algrorithm 1 returns an estimate $\hat{\boldsymbol{C}} = \boldsymbol{\Theta}_{\text{END}}$. Estimates of $\boldsymbol{A}$ and $\boldsymbol{B}$ are decoded from the left and right submatrices of $\hat{\boldsymbol{C}}$ respectively.

## 3 MAIN RESULTS

### 3.1 PRELIMINARIES

The analysis of the state equation naturally depends on the choice of the activation function; which is the source of nonlinearity. We first define a class of Lipschitz and increasing activation functions.

**Definition 3.1** ($\beta$-increasing activation). *Given* $1 \ge \beta \ge 0$, *the activation function* $\phi$ *satisfies* $\phi(0) = 0$ *and* $1 \ge \phi'(x) \ge \beta$ *for all* $x \in \mathbb{R}$.

Our results will apply to strictly increasing activations where $\phi$ is $\beta$-increasing for some $\beta > 0$. Observe that, this excludes ReLU activation which has zero derivative for negative values. However, it includes Leaky ReLU which is a generalization of ReLU. Parameterized by $1 \ge \beta \ge 0$, Leaky ReLU is a $\beta$-increasing function given by

$$\text{LReLU}(x) = \max(\beta x, x). \tag{3.1}$$

In general, given an increasing and 1-Lipschitz activation $\phi$, a $\beta$-increasing function $\phi_\beta$ can be obtained by blending $\phi$ with the linear activation, i.e. $\phi_\beta(x) = (1 - \beta)\phi(x) + \beta x$.

A critical property that enables SGD is that the state-vector covariance $\boldsymbol{\Sigma}[\boldsymbol{h}_t]$ is well-conditioned under proper assumptions. The lemma below provides upper and lower bounds on this covariance matrix in terms of problem variables.

**Lemma 3.2** (State vector covariance). *Consider the state equation* (1.1) *where* $\boldsymbol{h}_0 = 0$ *and* $\boldsymbol{u}_t \overset{i.i.d.}{\sim} \mathcal{N}(0, \boldsymbol{I}_p)$. *Define the upper bound term* $B_t$ *as*

$$B_t = \|\boldsymbol{B}\|\sqrt{\frac{1 - \|\boldsymbol{A}\|^{2t}}{1 - \|\boldsymbol{A}\|^2}}. \tag{3.2}$$

- *Suppose $\phi$ is 1-Lipschitz and $\phi(0) = 0$. Then, for all $t \geq 0$, $\boldsymbol{\Sigma}[\boldsymbol{h}_t] \preceq B_t^2 \boldsymbol{I}_n$.*

- *Suppose $\phi$ is a $\beta$-increasing function and $p \geq n$. Then, $\boldsymbol{\Sigma}[\boldsymbol{h}_t] \succeq \beta^2 s_{\min}(\boldsymbol{B})^2 \boldsymbol{I}_n$.*

As a natural extension from linear dynamical systems, we will say the system is stable if $\|\boldsymbol{A}\| < 1$ and unstable otherwise. For activations we consider, stability implies that if the input is set to 0, state vector $\boldsymbol{h}_t$ will exponentially converge to 0 i.e. the system forgets the past states quickly. This is also the reason $(B_t)_{t \geq 0}$ sequence converges for stable systems and diverges otherwise. The condition number of the covariance will play a critical role in our analysis. Using Lemma 3.2, this number can be upper bounded by $\rho$ defined as

$$\rho = \left(\frac{B_\infty}{\beta s_{\min}(\boldsymbol{B})}\right)^2 = \left(\frac{\|\boldsymbol{B}\|}{s_{\min}(\boldsymbol{B})}\right)^2 \frac{1}{\beta^2(1 - \|\boldsymbol{A}\|^2)}. \tag{3.3}$$

Observe that, the condition number of $\boldsymbol{B}$ appears inside the $\rho$ term.

## 3.2 Learning from Single Trajectory

Our main result applies to stable systems ($\|\boldsymbol{A}\| < 1$) and provides a non-asymptotic convergence guarantee for SGD in terms of the upper bound on the state vector covariance. This result characterizes the sample complexity and the rate of convergence of SGD; and also provides insights into the role of activation function and the spectral norm of $\boldsymbol{A}$.

**Theorem 3.3** (Main result). *Let $\{\boldsymbol{u}_t, \boldsymbol{h}_{t+1}\}_{t=1}^N$ be a finite trajectory generated from the state equation* (1.1). *Suppose $\|\boldsymbol{A}\| < 1$, $\phi$ is $\beta$-increasing, $\boldsymbol{h}_0 = 0$, $p \geq n$, and $\boldsymbol{u}_t \overset{i.i.d.}{\sim} \mathcal{N}(0, \boldsymbol{I}_p)$. Let $\rho$ be same as* (3.3) *and $c, C, c_0$ be properly chosen absolute constants. Pick the trajectory length $N$ to satisfy*
$$N \geq CL\rho^2(n + p),$$
*where $L = 1 - \frac{\log(cn\rho)}{\log\|\boldsymbol{A}\|}$. Pick scaling $\mu = 1/B_\infty$, learning rate $\eta = c_0 \frac{\beta^2}{\rho n(n+p)}$, and consider the loss function* (2.3). *With probability $1 - 4N\exp(-100n) - 8L\exp(-\mathcal{O}(\frac{N}{L\rho^2}))$, starting from an initial point $\boldsymbol{\Theta}_0$, for all $\tau \geq 0$, the SGD iterations described in Algorithm 1 satisfies*

$$\mathbb{E}[\|\boldsymbol{\Theta}_\tau - \boldsymbol{C}\|_F^2] \leq (1 - c_0 \frac{\beta^4}{2\rho^2 n(n+p)})^\tau \|\boldsymbol{\Theta}_0 - \boldsymbol{C}\|_F^2. \tag{3.4}$$

*Here the expectation is over the randomness of the SGD updates.*

**Sample complexity:** Theorem 3.3 essentially requires $N \gtrsim (n+p)/\beta^4$ samples for learning. This can be seen by unpacking (3.3) and ignoring the logarithmic $L$ term and the condition number of $\boldsymbol{B}$. Observe that $\mathcal{O}(n + p)$ growth achieves near-optimal sample size for our problem. Each state equation (1.1) consists of $n$ sub-equations (one for each entry of $\boldsymbol{h}_{t+1}$). We collect $N$ state equations to obtain a system of $Nn$ equations. On the other hand, the total number of unknown parameters in $\boldsymbol{A}$ and $\boldsymbol{B}$ are $n(n+p)$. This implies Theorem 3.3 is applicable as soon as the problem is mildly overdetermined i.e. $Nn \gtrsim n(n+p)$.

**Computational complexity:** Theorem 3.3 requires $\mathcal{O}(n(n + p)\log\frac{1}{\varepsilon})$ iterations to reach $\varepsilon$-neighborhood of the ground truth. Our analysis reveals that, this rate can be accelerated if the state vector is zero-mean. This happens for odd activation functions satisfying $\phi(-x) = -\phi(x)$ (e.g. linear activation). The result below is a corollary and requires $\times n$ less iterations.

**Theorem 3.4** (Faster learning for odd activations). *Consider the same setup provided in Theorem 3.3. Additionally, assume that $\phi$ is an odd function. Pick scaling $\mu = 1/B_\infty$, learning rate $\eta = c_0 \frac{\beta^2}{\rho(n+p)}$, and consider the loss function* (2.3). *With probability $1 - 4N\exp(-100n) - 8L\exp(-\mathcal{O}(\frac{N}{L\rho^2}))$, starting from an initial point $\boldsymbol{\Theta}_0$, for all $\tau \geq 0$, the SGD iterations described in Algorithm 1 satisfies*

$$\mathbb{E}[\|\boldsymbol{\Theta}_\tau - \boldsymbol{C}\|_F^2] \leq (1 - c_0 \frac{\beta^4}{2\rho^2(n+p)})^\tau \|\boldsymbol{\Theta}_0 - \boldsymbol{C}\|_F^2, \tag{3.5}$$

*where the expectation is over the randomness of the SGD updates.*

Another aspect of the convergence rate is the dependence on $\beta$. In terms of $\beta$, the SGD error (3.4) decays as $(1 - \mathcal{O}(\beta^8))^\tau$. While it is not clear how optimal is the exponent 8, numerical experiments in Section 6 demonstrate that larger $\beta$ indeed results in drastically faster convergence.

## 4    MAIN IDEAS AND PROOF STRATEGY

We first outline our high-level proof strategy for Theorem 3.3; which brings together ideas from statistics and optimization.

1. We first show that input data is well-behaved by proving that state-vector $\boldsymbol{h}_t$ has a well-conditioned covariance as discussed in Lemma 3.2 and shown in Appendix B. The key idea is if $\phi$ is $\beta$-increasing, then the random input data $\boldsymbol{u}_t$ provides sufficient excitation for the output state $\boldsymbol{h}_{t+1}$.

2. Even if individual samples are well-behaved, analyzing (2.3) is still challenging due to temporal dependencies between the samples. These dependencies prevent us from directly using statistical learning results that typically assume i.i.d. samples. We show that the dependency between samples at time $t$ and $t + T$ decay exponentially fast in separation $T$ (for stable systems). This is outlined in Appendix C.

3. This observation allows us to obtain nearly independent data by subsampling the original trajectory to get $(\boldsymbol{h}_{iT}, \boldsymbol{u}_{iT})_{i \geq 0}$. Thanks to exponential decay, a logarithmically small $T$ can be chosen to generate large subtrajectories of size $N/T$. Appendix D uses additional perturbation arguments to establish the well-behavedness of the overall data matrix.

4. To conclude, we obtain a deterministic result which establishes fast convergence result for $\beta$-increasing activations and well-behaved dataset. This is provided in Theorem 4.1 and proved in Appendix A.

The first three steps are related to the statistical nature of the problem which can be decoupled from the last step. Specifically, the last step derives a deterministic result that establishes the linear convergence of SGD for $\beta$-increasing functions. For linear convergence proofs, a typical strategy is showing the *strong convexity* of the loss function i.e. showing that, for some $\alpha > 0$ and all points $\boldsymbol{v}, \boldsymbol{u}$, the gradient satisfies

$$\langle \nabla \mathcal{L}(\boldsymbol{v}) - \nabla \mathcal{L}(\boldsymbol{u}), \boldsymbol{v} - \boldsymbol{u} \rangle \geq \alpha \|\boldsymbol{v} - \boldsymbol{u}\|_{\ell_2}^2.$$

The core idea of our convergence result is that the strong convexity parameter of the loss function with $\beta$-increasing activations can be connected to the loss function with *linear activations*. In particular, recalling (2.3), set $\boldsymbol{y}_t^{\mathrm{lin}} = \boldsymbol{C}\boldsymbol{x}_t$ and define the linear loss to be

$$\mathcal{L}^{\mathrm{lin}}(\boldsymbol{\Theta}) = \frac{1}{2N} \sum_{i=1}^{N} \|\boldsymbol{y}_t^{\mathrm{lin}} - \boldsymbol{\Theta}\boldsymbol{x}_t\|_{\ell_2}^2.$$

Denoting the strong convexity parameter of the original loss by $\alpha_\phi$ and that of linear loss by $\alpha_{\mathrm{lin}}$, we argue that $\alpha_\phi \geq \beta^2 \alpha_{\mathrm{lin}}$; which allows us to establish a convergence result as soon as $\alpha_{\mathrm{lin}}$ is strictly positive. Next result is our SGD convergence theorem which follows from this discussion.

**Theorem 4.1** (Deterministic convergence). *Suppose a data set $\{\boldsymbol{x}_i, \boldsymbol{y}_i\}_{i=1}^N$ is given; where output $\boldsymbol{y}_i$ is related to input $\boldsymbol{x}_i$ via $\boldsymbol{y}_i = \phi(\langle \boldsymbol{x}_i, \boldsymbol{\theta} \rangle)$ for some $\boldsymbol{\theta} \in \mathbb{R}^n$. Suppose $\beta > 0$ and $\phi$ is a $\beta$-increasing. Let $\gamma_+ \geq \gamma_- > 0$ be scalars. Assume that input samples satisfy the bounds*

$$\gamma_+ \boldsymbol{I}_n \succeq \frac{1}{N} \sum_{i=1}^{N} \boldsymbol{x}_i \boldsymbol{x}_i^T \succeq \gamma_- \boldsymbol{I}_n \quad , \quad \|\boldsymbol{x}_i\|_{\ell_2}^2 \leq B \text{ for all } i.$$

*Let $\{r_\tau\}_{\tau=0}^\infty$ be a sequence of i.i.d. integers uniformly distributed between $1$ to $N$. Then, starting from an arbitrary point $\boldsymbol{\theta}_0$, setting learning rate $\eta = \frac{\beta^2 \gamma_-}{\gamma_+ B}$, for all $\tau \geq 0$, the SGD iterations for quadratic loss*

$$\boldsymbol{\theta}_{\tau+1} = \boldsymbol{\theta}_\tau - \eta(\phi(\boldsymbol{x}_{r_\tau}^T \boldsymbol{\theta}_\tau) - \boldsymbol{y}_{r_\tau})\phi'(\boldsymbol{x}_{r_\tau}^T \boldsymbol{\theta}_\tau)\boldsymbol{x}_{r_\tau}, \tag{4.1}$$

*satisfies the error bound*

$$\mathbb{E}[\|\boldsymbol{\theta}_\tau - \boldsymbol{\theta}\|_{\ell_2}^2] \leq \|\boldsymbol{\theta}_0 - \boldsymbol{\theta}\|_{\ell_2}^2 (1 - \frac{\beta^4 \gamma_-^2}{\gamma_+ B})^\tau, \tag{4.2}$$

*where the expectation is over the random selection of the SGD iterations $\{r_\tau\}_{\tau=0}^\infty$.*

This theorem provides a clean convergence rate for SGD for $\beta$-increasing activations and naturally generalizes standard results on linear regression which corresponds to $\beta = 1$. We remark that related results appear in the literature on generalized linear models. Kakade et al. (2011); Foster et al. (2018); Mei et al. (2018a) provide learning theoretic loss/gradient/hessian convergence results for isotonic regression, robust regression, and $\beta$-increasing activations. Goel et al. (2018) establishes a similar result for leaky ReLU activations under the assumption of symmetric input distribution and infinitely many samples (i.e. in population limit). Compared to these, we establish a *deterministic* linear convergence guarantee for SGD that works whenever the data matrix is full rank. We believe extensions to proximal gradient methods might be beneficial for high-dimensional nonlinear problems (e.g. sparse/low-rank approximation, manifold constraints Cai et al. (2010); Beck & Teboulle (2009); Oymak et al. (2018); Agarwal et al. (2010); Pereira et al. (2010)) and is left as a future work.

To derive our main results in Section 3, we need to address the first three steps outlined earlier and determine the conditions under which Theorem 4.1 is applicable to the data obtained from RNN state equation with high probability. Below we provide desirable characteristics of the state vector; which enables our statistical results.

**Assumption 1** (Well-behaved state vector). *Let $L > 1$ be an integer. There exists positive scalars $\gamma_+, \gamma_-, \theta$ and an absolute constant $C > 0$ such that $\theta \leq 3\sqrt{n}$ and the following holds*

- **Lower bound:** $\Sigma[\boldsymbol{h}_{L-1}] \succeq \gamma_- \boldsymbol{I}_n$,

- **Upper bound:** *for all $t$, the state vector satisfies*

$$\Sigma[\boldsymbol{h}_t] \preceq \gamma_+ \boldsymbol{I}_n \quad , \quad \|\boldsymbol{h}_t - \mathbb{E}[\boldsymbol{h}_t]\|_{\psi_2} \leq C\sqrt{\gamma_+} \quad and \quad \|\mathbb{E}[\boldsymbol{h}_t]\|_{\ell_2} \leq \theta\sqrt{\gamma_+}. \tag{4.3}$$

    *Here $\|\cdot\|_{\psi_2}$ returns the subgaussian norm of a vector (see Def. 5.22 of Vershynin (2010)).*

Assumption 1 ensures that covariance is well-conditioned, state vector is well-concentrated, and it has a reasonably small expectation. Our next theorem establishes statistical guarantees for learning the RNN state equation based on this assumption.

**Theorem 4.2** (General result). *Let $\{\boldsymbol{u}_t, \boldsymbol{h}_{t+1}\}_{t=1}^N$ be a length $N$ trajectory of the state equation* (1.1). *Suppose $\|\boldsymbol{A}\| < 1$, $\phi$ is $\beta$-increasing, $\boldsymbol{h}_0 = 0$, and $\boldsymbol{u}_t \overset{i.i.d.}{\sim} \mathcal{N}(0, \boldsymbol{I}_p)$. Given scalars $\gamma_+ \geq \gamma_- > 0$, set the condition number as $\rho = \gamma_+/\gamma_-$. For absolute constants $C, c, c_0 > 0$, choose trajectory length $N$ to satisfy*

$$N \geq CL\rho^2(n+p) \quad where \quad L = \lceil 1 - \frac{\log(cn\rho)}{\log\|\boldsymbol{A}\|}\rceil.$$

*Suppose Assumption 1 holds with $L, \gamma_+, \gamma_-, \theta$. Pick scaling to be $\mu = 1/\sqrt{\gamma_+}$ and learning rate to be $\eta = c_0 \frac{\beta^2}{\rho(\theta+\sqrt{2})^2(n+p)}$. With probability $1 - 4N\exp(-100n) - 8L\exp(-\mathcal{O}(\frac{N}{L\rho^2}))$, starting from $\boldsymbol{\Theta}_0$, for all $\tau \geq 0$, the SGD iterations on loss* (2.3) *as described in Algorithm 1 satisfies*

$$\mathbb{E}[\|\boldsymbol{\Theta}_\tau - \boldsymbol{C}\|_F^2] \leq (1 - c_0 \frac{\beta^4}{2\rho^2(\theta+\sqrt{2})^2(n+p)})^\tau \|\boldsymbol{\Theta}_0 - \boldsymbol{C}\|_F^2, \tag{4.4}$$

*where the expectation is over the randomness of SGD updates.*

The advantage of this theorem is that, it isolates the optimization problem from the statistical properties of state vector. If one can prove tighter bounds on achievable $(\gamma_+, \gamma_-, \theta)$, it will immediately imply improved performance for SGD. In particular, Theorems 3.3 and 3.4 are simple corollaries of Theorem 4.2 with proper choices.

- Theorem 3.3 follows by setting $\gamma_+ = B_\infty^2$, $\gamma_- = \beta^2 s_{\min}(\boldsymbol{B})^2$, and $\theta = \sqrt{n}$.
- Theorem 3.4 follows by setting $\gamma_+ = B_\infty^2$, $\gamma_- = \beta^2 s_{\min}(\boldsymbol{B})^2$, and $\theta = 0$.

## 5 LEARNING UNSTABLE SYSTEMS

So far, we considered learning from a single RNN trajectory for stable systems ($\|\boldsymbol{A}\| < 1$). For such systems, as the time goes on, the impact of the earlier states disappear. In our analysis, this allows us to split a single trajectory into multiple nearly-independent trajectories. This approach will not work for unstable systems ($\boldsymbol{A}$ is arbitrary) where the impact of older states may be amplified over time. To address this, we consider a model where the data is sampled from multiple independent trajectories.

Suppose $N$ independent trajectories of the state-equation (1.1) are available. Pick some integer $T_0 \geq 1$. Denoting the $i$th trajectory by the triple $(\boldsymbol{h}_{t+1}^{(i)}, \boldsymbol{h}_t^{(i)}, \boldsymbol{u}_t^{(i)})_{t \geq 0}$, we collect a single sample from each trajectory at time $T_0$ to obtain the triple $(\boldsymbol{h}_{T_0+1}^{(i)}, \boldsymbol{h}_{T_0}^{(i)}, \boldsymbol{u}_{T_0}^{(i)})$. To utilize the existing optimization framework (2.3); for $1 \leq i \leq N$, we set,

$$(\boldsymbol{y}_i, \boldsymbol{h}_i, \boldsymbol{u}_i) = (\boldsymbol{h}_{T_0+1}^{(i)}, \boldsymbol{h}_{T_0}^{(i)}, \boldsymbol{u}_{T_0}^{(i)}). \tag{5.1}$$

With this setup, we can again use the SGD Algorithm 1 to learn the weights $\boldsymbol{A}$ and $\boldsymbol{B}$. The crucial difference compared to Section 3 is that, the samples $(\boldsymbol{y}_i, \boldsymbol{h}_i, \boldsymbol{u}_i)_{i=1}^N$ are now independent of each other; hence, the analysis is simplified. As previously, having an upper bound on the condition number of the state-vector covariance is critical. This upper bound can be shown to be $\rho$ defined as

$$\rho = \begin{cases} \bar{\rho} & \text{if } n > 1 \\ \bar{\rho} \frac{1 - \beta^2 |\boldsymbol{A}|^2}{1 - (\beta|\boldsymbol{A}|)^{2T_0}} & \text{if } n = 1 \end{cases} \quad \text{where} \quad \bar{\rho} = \frac{B_{T_0}^2}{\beta^2 s_{\min}(\boldsymbol{B})^2}. \tag{5.2}$$

The $\bar{\rho}$ term is similar to the earlier definition (3.3); however it involves $B_{T_0}$ rather than $B_\infty$. This modification is indeed necessary since $B_\infty = \infty$ when $\|\boldsymbol{A}\| > 1$. On the other hand, note that, $B_{T_0}^2$ grows proportional to $\|\boldsymbol{A}\|^{2T_0}$; which results in exponentially bad condition number in $T_0$. Our $\rho$ definition remedies this issue for single-output systems; where $n = 1$ and $\boldsymbol{A}$ is a scalar. In particular, when $\beta = 1$ (e.g. $\phi$ is linear) $\rho$ becomes equal to the correct value $1^2$. The next theorem provides our result on unstable systems in terms of this condition number and other model parameters.

**Theorem 5.1** (Unstable systems). *Suppose we are given $N$ independent trajectories $(\boldsymbol{h}_t^{(i)}, \boldsymbol{u}_t^{(i)})_{t \geq 0}$ for $1 \leq i \leq N$. Each trajectory is sampled at time $T_0$ to obtain $N$ samples $(\boldsymbol{y}_i, \boldsymbol{h}_i, \boldsymbol{u}_i)_{i=1}^N$ where the $i$th sample is given by (5.1). Suppose the sample size satisfies*

$$N \geq C\rho^2(n + p)$$

*where $\rho$ is given by (5.2). Assume the initial states are $0$, $\phi$ is $\beta$-increasing, $p \geq n$, and $\boldsymbol{u}_t \overset{i.i.d.}{\sim} \mathcal{N}(0, \boldsymbol{I}_p)$. Set scaling $\mu = 1/\sqrt{B_{T_0}}$, learning rate $\eta = c_0 \frac{\beta^2}{\rho n(n+p)}$, and run SGD over the equations described in (2.2) and (2.3). Starting from $\boldsymbol{\Theta}_0$, with probability $1 - 2N \exp(-100(n+p)) - 4\exp(-\mathcal{O}(\frac{N}{\rho^2}))$, all SGD iterations satisfy*

$$\mathbb{E}[\|\boldsymbol{\Theta}_\tau - \boldsymbol{C}\|_F^2] \leq (1 - c_0 \frac{\beta^4}{2\rho^2 n(n+p)})^\tau \|\boldsymbol{\Theta}_0 - \boldsymbol{C}\|_F^2,$$

*where the expectation is over the randomness of the SGD updates.*

## 6 NUMERICAL EXPERIMENTS

We conducted experiments on ReLU and Leaky ReLU activations. Let us first describe the experimental setup. We pick the state dimension $n = 50$ and the input dimension $p = 100$. We choose the ground truth matrix $\boldsymbol{A}$ to be a scaled random unitary matrix; which ensures that all singular values of $\boldsymbol{A}$ are equal. $\boldsymbol{B}$ is generated with i.i.d. $\mathcal{N}(0, 1)$ entries. Instead of using the theoretical scaling choice, we determine the scaling $\mu$ from empirical covariance matrices outlined in Algorithm 2. Similar to our proof strategy, this algorithm equalizes the spectral norms of the input and state covariances to speed up convergence. We also empirically determined the learning rate and used $\eta = 1/100$ in all experiments.

---

[2] Clearly, any nonzero $1 \times 1$ covariance matrix has condition number 1. However, due to subtleties in the proof strategy, we don't use $\rho = 1$ for $\beta < 1$. Obtaining tighter bounds on the subgaussian norm of the state-vector would help resolve this issue.

---

**Algorithm 2** Empirical hyperparameter selection.

---

1: **Inputs:** $(h_t, u_t)_{t=1}^N$ sampled from a trajectory.
2: **Outputs:** Scaling $\mu$.
3: Form the empirical covariance matrix $\Sigma_h$ from $\{h_t\}_{t=1}^N$.
4: Form the empirical covariance matrix $\Sigma_u$ from $\{u_t\}_{t=1}^N$.
5: **return** $\sqrt{\|\Sigma_u\|/\|\Sigma_h\|}$.

---

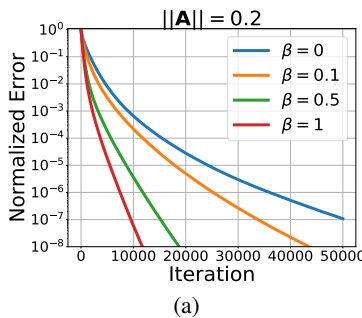 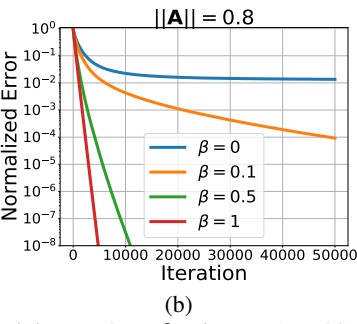

Figure 1: SGD convergence behavior for Leaky ReLUs with varying minimum slope $\beta$. Figures a) and b) repeat the same experiments. The difference is the spectral norm of the ground truth state matrix $\boldsymbol{A}$.

**Evaluation:** We consider two performance measures in the experiments. Let $\hat{\boldsymbol{C}}$ be an estimate of the ground truth parameter $\boldsymbol{C} = [\mu^{-1}\boldsymbol{A}\ \boldsymbol{B}]$. The first measure is the normalized error defined as $\|\hat{\boldsymbol{C}} - \boldsymbol{C}\|_F^2/\|\boldsymbol{C}\|_F^2$. The second measure is the normalized loss defined as

$$\frac{\sum_{i=1}^N \|\boldsymbol{y}_t - \phi(\hat{\boldsymbol{C}}\boldsymbol{x}_t)\|_{\ell_2}^2}{\sum_{i=1}^N \|\boldsymbol{y}_t\|_{\ell_2}^2}.$$

In all experiments, we run Algorithm 1 for 50000 SGD iterations and plot these measures as a function of $\tau$; by using the estimate available at the end of the $\tau$th SGD iteration for $0 \le \tau \le 50000$. Each curve is obtained by averaging the outcomes of 20 independent realizations. Our first experiments use $N = 500$; which is mildly larger than the total dimension $n + p = 150$. In Figure 1, we plot the Leaky ReLU errors with varying slopes as described in (3.1). Here $\beta = 0$ corresponds to ReLU and $\beta = 1$ is the linear model. In consistence with our theory, SGD achieves linear convergence and as $\beta$ increases, the rate of convergence drastically improves[3]. The improvement is more visible for less stable systems driven by $\boldsymbol{A}$ with a larger spectral norm. In particular, while ReLU converges for small $\|\boldsymbol{A}\|$, SGD gets stuck before reaching the ground truth when $\|\boldsymbol{A}\| = 0.8$.

To understand, how well SGD fits the training data, in Figure 2a, we plotted the normalized loss for ReLU activation. For more unstable system ($\|\boldsymbol{A}\| = 0.9$), training loss stagnates in a similar fashion to the parameter error. We also verified that the norm of the overall gradient $\|\nabla\mathcal{L}(\Theta_\tau)\|_F$ continues to decay (where $\Theta_\tau$ is the $\tau$th SGD iterate); which implies that SGD converges before reaching a global minima. As $\boldsymbol{A}$ becomes more stable, rate of convergence improves and linear rate is visible. Finally, to better understand the population landscape of the quadratic loss with ReLU activations, Figure 2b repeats the same ReLU experiments while increasing the sample size five times to $N = 2500$. For this more overdetermined problem, SGD converges even for $\|\boldsymbol{A}\| = 0.9$; indicating that

- population landscape of loss with ReLU activation is well-behaved,
- however ReLU problem requires more data compared to the Leaky ReLU for finding global minima.

Overall, as predicted by our theory, experiments verify that SGD indeed quickly finds the optimal weight matrices of the state equation (1.1) and as the activation slope $\beta$ increases, the convergence rate improves.

---

[3]Note that convergence becomes faster for larger $\beta$ under the realizable model i.e. there exists a ground truth state equation with activation slope $\beta$ that can fit the observed trajectory. This is consistent with the technical setup our results are proven. Also note that the data distribution in the experiments changes with the activation slope $\beta$. If the dataset is fixed and not realizable, the results may be different as we vary the slope $\beta$.

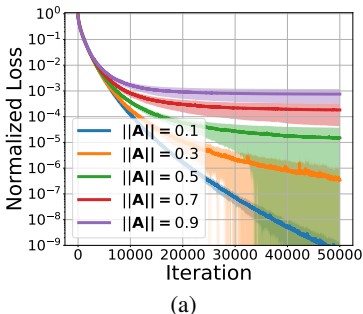 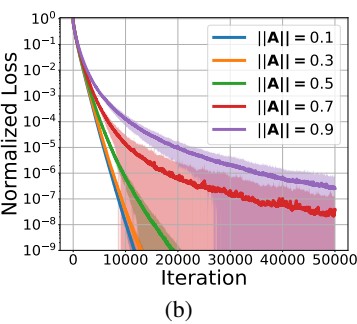

|(a)|(b)|

Figure 2: SGD convergence behavior for ReLU with varying spectral norm of the state matrix $\boldsymbol{A}$. Figures a) and b) repeats the same experiments. The difference is that a) uses $N = 500$ trajectory length whereas b) uses $N = 2500$ (i.e. $\times 5$ more data). Shaded regions highlight the one standard deviation around the mean.

## 7 CONCLUSIONS

This work showed that SGD can learn the nonlinear dynamical system (1.1); which is characterized by weight matrices and an activation function. This problem is of interest for recurrent neural networks as well as nonlinear system identification. We showed that efficient learning is possible with optimal sample complexity and good computational performance. Our results apply to strictly increasing activations such as Leaky ReLU. We empirically showed that Leaky ReLU converges faster than ReLU and requires less samples; in consistence with our theory. We list a few unanswered problems that would provide further insights into recurrent neural networks.

• **Covariance of the state-vector:** Our results depend on the covariance of the state-vector and requires it to be positive definite. One might be able to improve the current bounds on the condition number and relax the assumptions on the activation function. Deriving similar performance bounds for ReLU is particularly interesting.

• **Hidden state:** For RNNs, the state vector is hidden and is observed through an additional equation (2.1); which further complicates the optimization landscape. Even for linear dynamical systems, learning the $(\boldsymbol{A}, \boldsymbol{B}, \boldsymbol{C}, \boldsymbol{D})$ system ((1.1), (2.1)) is a non-trivial task Ho & Kalman (1966); Hardt et al. (2016). What can be said when we add the nonlinear activations?

• **Classification task:** In this work, we used normally distributed input and least-squares regression for our theoretical guarantees. More realistic input distributions might provide better insight into contemporary problems, such as natural language processing; where the goal is closer to classification (e.g. finding the best translation from another language).

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

# A    DETERMINISTIC CONVERGENCE RESULT FOR SGD

*Proof of Theorem 4.1.* Given two distinct scalars $a, b$; define $\phi'(a,b) = \frac{\phi(a)-\phi(b)}{a-b}$. $\phi'(a,b) \geq \beta$ since $\phi$ is $\beta$-increasing. Define $\boldsymbol{w}_\tau$ to be the residual $\boldsymbol{w}_\tau = \boldsymbol{\theta}_\tau - \boldsymbol{\theta}$. Observing

$$\phi(\boldsymbol{x}_{r_\tau}^T \boldsymbol{\theta}_\tau) - \boldsymbol{y}_{r_\tau} = \phi'(\boldsymbol{x}_{r_\tau}^T \boldsymbol{\theta}_\tau, \boldsymbol{x}_{r_\tau}^T \boldsymbol{\theta}) \boldsymbol{x}_{r_\tau}^T \boldsymbol{w}_\tau,$$

the SGD recursion obeys

$$\|\boldsymbol{w}_{\tau+1}\|_{\ell_2}^2 = \|\boldsymbol{w}_\tau - \eta(\phi(\boldsymbol{x}_{r_\tau}^T \boldsymbol{\theta}_\tau) - \boldsymbol{y}_{r_\tau})\phi'(\boldsymbol{x}_{r_\tau}^T \boldsymbol{\theta}_\tau)\boldsymbol{x}_{r_\tau}\|_{\ell_2}^2.$$
$$= \|\boldsymbol{w}_\tau - \eta\boldsymbol{x}_{r_\tau}\phi'(\boldsymbol{x}_{r_\tau}^T \boldsymbol{\theta}_\tau)\phi'(\boldsymbol{x}_{r_\tau}^T \boldsymbol{\theta}_\tau, \boldsymbol{x}_{r_\tau}^T \boldsymbol{\theta})\boldsymbol{x}_{r_\tau}^T \boldsymbol{w}_\tau\|_{\ell_2}^2$$
$$= \|(\boldsymbol{I} - \eta\boldsymbol{G}_{r_\tau})\boldsymbol{w}_\tau\|_{\ell_2}^2$$

where $\boldsymbol{G}_{r_\tau} = \boldsymbol{x}_{r_\tau}\phi'(\boldsymbol{x}_{r_\tau}^T \boldsymbol{\theta}_\tau)\phi'(\boldsymbol{x}_{r_\tau}^T \boldsymbol{\theta}_\tau, \boldsymbol{x}_{r_\tau}^T \boldsymbol{\theta})\boldsymbol{x}_{r_\tau}^T$. Since $\phi$ is 1-Lipschitz and $\beta$-increasing, $\boldsymbol{G}_{r_\tau}$ is a positive-semidefinite matrix satisfying

$$\boldsymbol{x}_{r_\tau}\boldsymbol{x}_{r_\tau}^T \succeq \boldsymbol{G}_{r_\tau} \succeq \beta^2 \boldsymbol{x}_{r_\tau}\boldsymbol{x}_{r_\tau}^T,$$
$$\boldsymbol{G}_{r_\tau}^T \boldsymbol{G}_{r_\tau} \preceq \boldsymbol{x}_{r_\tau}\boldsymbol{x}_{r_\tau}^T \boldsymbol{x}_{r_\tau}\boldsymbol{x}_{r_\tau}^T \preceq B\boldsymbol{x}_{r_\tau}\boldsymbol{x}_{r_\tau}^T.$$

Consequently, we find the following bounds in expectation

$$\gamma_+ \boldsymbol{I}_n \succeq \mathbb{E}[\boldsymbol{G}_{r_\tau}] \succeq \beta^2 \gamma_- \boldsymbol{I}_n, \tag{A.1}$$
$$\mathbb{E}[\boldsymbol{G}_{r_\tau}^T \boldsymbol{G}_{r_\tau}] \preceq B\gamma_+ \boldsymbol{I}_n.$$

Observe that (A.1) essentially lower bounds the *strong convexity* parameter of the problem with $\beta^2 \gamma_-$; which is the strong convexity of the identical problem with the linear activation (i.e. $\beta = 1$). However, we only consider strong convexity around the ground truth parameter $\boldsymbol{\theta}$ i.e. we restricted our attention to $(\boldsymbol{\theta}, \boldsymbol{\theta}_\tau)$ pairs. With this, $\boldsymbol{w}_{\tau+1}$ can be controlled as,

$$\mathbb{E}[\|\boldsymbol{w}_{\tau+1}\|_{\ell_2}^2] = \mathbb{E}[\|(\boldsymbol{I} - \eta\boldsymbol{G}_{r_\tau})\boldsymbol{w}_\tau\|_{\ell_2}^2]$$
$$= \|\boldsymbol{w}_\tau\|_{\ell_2}^2 - 2\eta\,\mathbb{E}[\boldsymbol{w}_\tau^T \boldsymbol{G}_{r_\tau} \boldsymbol{w}_\tau] + \eta^2\,\mathbb{E}[\boldsymbol{w}_\tau^T \boldsymbol{G}_{r_\tau}^T \boldsymbol{G}_{r_\tau} \boldsymbol{w}_\tau]$$
$$\leq \|\boldsymbol{w}_\tau\|_{\ell_2}^2 (1 - 2\eta\beta^2\gamma_- + \eta^2 B\gamma_+).$$

Setting $\eta = \frac{\beta^2 \gamma_-}{\gamma_+ B}$, we find the advertised bound

$$\mathbb{E}[\|\boldsymbol{w}_{\tau+1}\|_{\ell_2}^2] \leq \mathbb{E}[\|\boldsymbol{w}_\tau\|_{\ell_2}^2](1 - \frac{\beta^4 \gamma_-^2}{\gamma_+ B}).$$

Applying induction over the iterations $\tau$, we find the advertised bound (4.2)

$$\mathbb{E}[\|\boldsymbol{w}_\tau\|_{\ell_2}^2] \leq \|\boldsymbol{w}_0\|_{\ell_2}^2 (1 - \frac{\beta^4 \gamma_-^2}{\gamma_+ B})^\tau.$$

$\square$

**Lemma A.1** (Merging $L$ splits). *Assume matrices $\boldsymbol{X}^{(i)} \in \mathbb{R}^{N_i \times q}$ are given for $1 \leq i \leq L$. Suppose for all $1 \leq i \leq L$, rows of $\boldsymbol{X}^{(i)}$ has $\ell_2$ norm at most $\sqrt{B}$ and each $\boldsymbol{X}^{(i)}$ satisfies*

$$\gamma_+ \boldsymbol{I}_n \succeq \frac{\boldsymbol{X}^{(i)^T} \boldsymbol{X}^{(i)}}{N_i} \succeq \gamma_- \boldsymbol{I}_n.$$

*Set $N = \sum_{i=1}^{L} N_i$ and form the concatenated matrix $\boldsymbol{X} = \begin{bmatrix} \boldsymbol{X}^{(1)} \\ \boldsymbol{X}^{(2)} \\ \vdots \\ \boldsymbol{X}^{(L)} \end{bmatrix}$. Denote $i$th row of $\boldsymbol{X}$ by $\boldsymbol{x}_i$. Then, for each $i$, $\|\boldsymbol{x}_i\|_{\ell_2}^2 \leq B$ and*

$$\gamma_+ \boldsymbol{I}_n \succeq \frac{\boldsymbol{X}^T \boldsymbol{X}}{N} = \frac{1}{N}\sum_{i=1}^{N} \boldsymbol{x}_i \boldsymbol{x}_i^T \succeq \gamma_- \boldsymbol{I}_n.$$

*Proof.* The bound on the rows $\|\boldsymbol{x}_i\|_{\ell_2}$ directly follows by assumption. For the remaining result, first observe that $\boldsymbol{X}^T \boldsymbol{X} = \sum_{i=1}^{L} \boldsymbol{X}^{(i)^T} \boldsymbol{X}^{(i)}$. Next, we have

$$N\gamma_+ \boldsymbol{I}_n = \sum_{i=1}^{L} N_i \gamma_+ \boldsymbol{I}_n \succeq \sum_{i=1}^{L} \boldsymbol{X}^{(i)^T} \boldsymbol{X}^{(i)} \succeq \sum_{i=1}^{L} N_i \gamma_- \boldsymbol{I}_n = N\gamma_- \boldsymbol{I}_n.$$

Combining these two yields the desired upper/lower bounds on $\boldsymbol{X}^T \boldsymbol{X}/N$. $\square$

# B  PROPERTIES OF THE NONLINEAR STATE EQUATIONS

This section characterizes the properties of the state vector $\boldsymbol{h}_t$ when input sequence is normally distributed. These bounds will be crucial for obtaining upper and lower bounds for the singular values of the data matrix $\boldsymbol{X} = [\boldsymbol{x}_1 \ \dots \ \boldsymbol{x}_N]^T$ described in (2.2). For probabilistic arguments, we will use the properties of subgaussian random variables. Orlicz norm provides a general framework that subsumes subgaussianity.

**Definition B.1** (Orlicz norms). *For a scalar random variable Orlicz-a norm is defined as*

$$\|X\|_{\psi_a} = \sup_{k \geq 1} k^{-1/a} (\mathbb{E}[|X|^k])^{1/k}$$

*Orlicz-a norm of a vector $\boldsymbol{x} \in \mathbb{R}^p$ is defined as $\|\boldsymbol{x}\|_{\psi_a} = \sup_{\boldsymbol{v} \in \mathcal{B}^p} \|\boldsymbol{v}^T \boldsymbol{x}\|_{\psi_a}$ where $\mathcal{B}^p$ is the unit $\ell_2$ ball. The subexponential norm is the Orlicz-1 norm $\|\cdot\|_{\psi_1}$ and the subgaussian norm is the Orlicz-2 norm $\|\cdot\|_{\psi_2}$.*

**Lemma B.2** (Lipschitz properties of the state vector). *Consider the state equation* (1.1). *Suppose activation $\phi$ is 1-Lipschitz. Observe that $\boldsymbol{h}_{t+1}$ is a deterministic function of the input sequence $\{\boldsymbol{u}_\tau\}_{\tau=0}^t$. Fixing all vectors $\{\boldsymbol{u}_i\}_{i \neq \tau}$ (i.e. all except $\boldsymbol{u}_\tau$), $\boldsymbol{h}_{t+1}$ is $\|\boldsymbol{A}\|^{t-\tau}\|\boldsymbol{B}\|$ Lipschitz function of $\boldsymbol{u}_\tau$ for $0 \leq \tau \leq t$.*

*Proof.* Fixing $\{\boldsymbol{u}_i\}_{i \neq \tau}$, denote $\boldsymbol{h}_{t+1}$ as a function of $\boldsymbol{u}_\tau$ by $\boldsymbol{h}_{t+1}(\boldsymbol{u}_\tau)$. Given a pair of vectors $\boldsymbol{u}_\tau, \boldsymbol{u}_\tau'$ using 1-Lipschitzness of $\phi$, for any $t > \tau$, we have

$$\begin{aligned}
\|\boldsymbol{h}_{t+1}(\boldsymbol{u}_\tau) - \boldsymbol{h}_{t+1}(\boldsymbol{u}_\tau')\|_{\ell_2} &\leq \|\phi(\boldsymbol{A}\boldsymbol{h}_t(\boldsymbol{u}_\tau) + \boldsymbol{B}\boldsymbol{u}_t) - \phi(\boldsymbol{A}\boldsymbol{h}_t(\boldsymbol{u}_\tau') + \boldsymbol{B}\boldsymbol{u}_t)\|_{\ell_2} \\
&\leq \|\boldsymbol{A}(\boldsymbol{h}_t(\boldsymbol{u}_\tau) - \boldsymbol{h}_t(\boldsymbol{u}_\tau'))\|_{\ell_2} \\
&\leq \|\boldsymbol{A}\|\|\boldsymbol{h}_t(\boldsymbol{u}_\tau) - \boldsymbol{h}_t(\boldsymbol{u}_\tau')\|_{\ell_2}.
\end{aligned}$$

Proceeding with this recursion until $t = \tau$, we find

$$\begin{aligned}
\|\boldsymbol{h}_{t+1}(\boldsymbol{u}_\tau) - \boldsymbol{h}_{t+1}(\boldsymbol{u}_\tau')\|_{\ell_2} &\leq \|\boldsymbol{A}\|^{t-\tau}\|\boldsymbol{h}_{\tau+1}(\boldsymbol{u}_\tau) - \boldsymbol{h}_{\tau+1}(\boldsymbol{u}_\tau')\|_{\ell_2} \\
&\leq \|\boldsymbol{A}\|^{t-\tau}\|\phi(\boldsymbol{A}\boldsymbol{h}_\tau + \boldsymbol{B}\boldsymbol{u}_\tau) - \phi(\boldsymbol{A}\boldsymbol{h}_\tau + \boldsymbol{B}\boldsymbol{u}_\tau')\|_{\ell_2} \\
&\leq \|\boldsymbol{A}\|^{t-\tau}\|\boldsymbol{B}\|\|\boldsymbol{u}_\tau - \boldsymbol{u}_\tau'\|_{\ell_2}.
\end{aligned}$$

This bound implies $\boldsymbol{h}_{t+1}(\boldsymbol{u}_\tau)$ is $\|\boldsymbol{A}\|^{t-\tau}\|\boldsymbol{B}\|$ Lipschitz function of $\boldsymbol{u}_\tau$. □

**Lemma B.3** (Upper bound). *Consider the state equation governed by equation* (1.1). *Suppose $\boldsymbol{u}_t \overset{i.i.d.}{\sim} \mathcal{N}(0, \boldsymbol{I}_p)$, $\phi$ is 1-Lipschitz, $\phi(0) = 0$ and $\boldsymbol{h}_0 = 0$. Recall the definition* (3.2) *of $B_t$. We have the following properties*

- *$\boldsymbol{h}_t$ is a $B_t$-Lipschitz function of the vector $\boldsymbol{q}_t = [\boldsymbol{u}_0^T \ \dots \ \boldsymbol{u}_{t-1}^T]^T \in \mathbb{R}^{tp}$.*

- *There exists an absolute constant $c > 0$ such that $\|\boldsymbol{h}_t - \mathbb{E}[\boldsymbol{h}_t]\|_{\psi_2} \leq cB_t$ and $\boldsymbol{\Sigma}[\boldsymbol{h}_t] \preceq B_t^2 \boldsymbol{I}_n$.*

- *$\boldsymbol{h}_t$ satisfies*

$$\mathbb{E}[\|\boldsymbol{h}_t\|_{\ell_2}^2] \leq tr(\boldsymbol{B}\boldsymbol{B}^T)\frac{1 - \|\boldsymbol{A}\|^{2t}}{1 - \|\boldsymbol{A}\|^2} \leq \min\{n, p\}B_t^2.$$

  *Also, there exists an absolute constant $c > 0$ such that for any $m \geq n$, with probability $1 - 2\exp(-100m)$, $\|\boldsymbol{h}_t\|_{\ell_2} \leq c\sqrt{m}B_t$.*

*Proof.* **i) Bounding Lipschitz constant:** Observe that $\boldsymbol{h}_t$ is a deterministic function of $\boldsymbol{q}_t$ i.e. $\boldsymbol{h}_t = f(\boldsymbol{q}_t)$ for some function $f$. To bound Lipschitz constant of $f$, for all (deterministic) vector pairs $\boldsymbol{q}_t$ and $\hat{\boldsymbol{q}}_t$, we find a scalar $L_f$ satisfying,

$$\|f(\boldsymbol{q}_t) - f(\hat{\boldsymbol{q}}_t)\|_{\ell_2} \leq L_f \|\boldsymbol{q}_t - \hat{\boldsymbol{q}}_t\|_{\ell_2}. \tag{B.1}$$

Define the vectors, $\{\boldsymbol{a}_i\}_{i=0}^t$, as follows

$$\boldsymbol{a}_i = [\hat{\boldsymbol{u}}_0^T \ \dots \ \hat{\boldsymbol{u}}_{i-1}^T \ \boldsymbol{u}_i^T \ \dots \ \boldsymbol{u}_{t-1}^T]^T.$$

Observing that $\boldsymbol{a}_0 = \boldsymbol{q}_t$, $\boldsymbol{a}_t = \hat{\boldsymbol{q}}_t$, we write the telescopic sum,

$$\|f(\boldsymbol{q}_t) - f(\hat{\boldsymbol{q}}_t)\|_{\ell_2} \leq \sum_{i=0}^{t-1} \|f(\boldsymbol{a}_{i+1}) - f(\boldsymbol{a}_i)\|_{\ell_2}.$$

Focusing on the individual terms $f(\boldsymbol{a}_{i+1}) - f(\boldsymbol{a}_i)$, observe that the only difference is the $\boldsymbol{u}_i, \hat{\boldsymbol{u}}_i$ terms. Viewing $\boldsymbol{h}_t$ as a function of $\boldsymbol{u}_i$ and applying Lemma B.2,

$$\|f(\boldsymbol{a}_{i+1}) - f(\boldsymbol{a}_i)\|_{\ell_2} \leq \|\boldsymbol{A}\|^{t-1-i}\|\boldsymbol{B}\|\|\boldsymbol{u}_i - \hat{\boldsymbol{u}}_i\|_{\ell_2}.$$

To bound the sum, we apply the Cauchy-Schwarz inequality; which yields

$$
\begin{aligned}
|f(\boldsymbol{q}_t) - f(\hat{\boldsymbol{q}}_t)| &\leq \sum_{i=0}^{t-1} \|\boldsymbol{A}\|^{t-1-i} \|\boldsymbol{B}\| \|\boldsymbol{u}_i - \hat{\boldsymbol{u}}_i\|_{\ell_2} \\
&\leq (\sum_{i=0}^{t-1} \|\boldsymbol{A}\|^{2(t-1-i)} \|\boldsymbol{B}\|^2)^{1/2} \underbrace{(\sum_{i=0}^{t-1} \|\boldsymbol{u}_i - \hat{\boldsymbol{u}}_i\|_{\ell_2}^2)^{1/2}}_{\|\boldsymbol{q}_t - \hat{\boldsymbol{q}}_t\|_{\ell_2}} \\
&\leq \sqrt{\frac{\|\boldsymbol{B}\|^2 (1 - \|\boldsymbol{A}\|^{2t})}{1 - \|\boldsymbol{A}\|^2}} \|\boldsymbol{q}_t - \hat{\boldsymbol{q}}_t\|_{\ell_2} \\
&= B_t \|\boldsymbol{q}_t - \hat{\boldsymbol{q}}_t\|_{\ell_2}.
\end{aligned}
\tag{B.2}
$$

The final line achieves the inequality (B.1) with $L_f = B_t$ hence $\boldsymbol{h}_t$ is $B_t$ Lipschitz function of $\boldsymbol{q}_t$.

**ii) Bounding subgaussian norm:** When $\boldsymbol{u}_t \overset{\text{i.i.d.}}{\sim} \mathcal{N}(0, \boldsymbol{I}_p)$, the vector $\boldsymbol{q}_t$ is distributed as $\mathcal{N}(0, \boldsymbol{I}_{tp})$. Since $\boldsymbol{h}_t$ a $B_t$ Lipschitz function of $\boldsymbol{q}_t$, for any fixed unit length vector $\boldsymbol{v}$, $\alpha_{\boldsymbol{v}} := \boldsymbol{v}^T \boldsymbol{h}_t = \boldsymbol{v}^T f(\boldsymbol{q}_t)$ is still $B_t$-Lipschitz function of $\boldsymbol{q}_t$. Hence, using Gaussian concentration of Lipschitz functions, $\alpha_{\boldsymbol{v}}$ satisfies

$$
\mathbb{P}(|\alpha_{\boldsymbol{v}} - \mathbb{E}[\alpha_{\boldsymbol{v}}]| \geq t) \leq 2\exp(-\frac{t^2}{2B_t^2}).
$$

This implies that for any $\boldsymbol{v}$, $\alpha_{\boldsymbol{v}} - \mathbb{E}[\alpha_{\boldsymbol{v}}]$ is $\mathcal{O}(B_t)$ subgaussian Vershynin (2010). This is true for all unit $\boldsymbol{v}$, hence using Definition B.1, the vector $\boldsymbol{h}_t$ satisfies $\|\boldsymbol{h}_t - \mathbb{E}[\boldsymbol{h}_t]\|_{\psi_2} \leq \mathcal{O}(B_t)$ as well. Secondly, $B_t$-Lipschitz function of a Gaussian vector obeys the variance inequality $\mathbf{var}[\alpha_{\boldsymbol{v}}] \leq B_t^2$ (page 49 of Ledoux (2001)), which implies the covariance bound

$$
\boldsymbol{\Sigma}[\boldsymbol{h}_t] \preceq B_t^2 \boldsymbol{I}_n.
$$

**iii) Bounding $\ell_2$-norm:** To obtain this result, we first bound $\mathbb{E}[\|\boldsymbol{h}_t\|_{\ell_2}^2]$. Since $\phi$ is 1-Lipschitz and $\phi(0) = 0$, we have the deterministic relation

$$
\|\boldsymbol{h}_{t+1}\|_{\ell_2} \leq \|\boldsymbol{A}\boldsymbol{h}_t + \boldsymbol{B}\boldsymbol{u}_t\|_{\ell_2}.
$$

Taking squares of both sides, expanding the right hand side, and using the independence of $\boldsymbol{h}_t, \boldsymbol{u}_t$ and the covariance information of $\boldsymbol{u}_t$, we obtain

$$
\mathbb{E}[\|\boldsymbol{h}_{t+1}\|_{\ell_2}^2] \leq \mathbb{E}[\|\boldsymbol{A}\boldsymbol{h}_t + \boldsymbol{B}\boldsymbol{u}_t\|_{\ell_2}^2] = \mathbb{E}[\|\boldsymbol{A}\boldsymbol{h}_t\|_{\ell_2}^2] + \mathbb{E}[\|\boldsymbol{B}\boldsymbol{u}_t\|_{\ell_2}^2]
\tag{B.3}
$$

$$
\leq \|\boldsymbol{A}\|^2 \mathbb{E}[\|\boldsymbol{h}_t\|_{\ell_2}^2] + \mathrm{tr}(\boldsymbol{B}\boldsymbol{B}^T).
\tag{B.4}
$$

Now that the recursion is established, expanding $\boldsymbol{h}_t$ on the right hand side until $\boldsymbol{h}_0 = 0$, we obtain

$$
\mathbb{E}[\|\boldsymbol{h}_{t+1}\|_{\ell_2}^2] \leq \sum_{i=0}^{t} \|\boldsymbol{A}\|^{2i} \mathrm{tr}(\boldsymbol{B}\boldsymbol{B}^T) \leq \mathrm{tr}(\boldsymbol{B}\boldsymbol{B}^T) \frac{1 - \|\boldsymbol{A}\|^{2(t+1)}}{1 - \|\boldsymbol{A}\|^2}.
$$

Now using the fact that $\mathrm{tr}(\boldsymbol{B}\boldsymbol{B}^T) \leq \mathrm{rank}(\boldsymbol{B}) \|\boldsymbol{B}\|^2 \leq \min\{n, p\} \|\boldsymbol{B}\|^2$, we find

$$
\mathbb{E}[\|\boldsymbol{h}_{t+1}\|_{\ell_2}]^2 \leq \mathbb{E}[\|\boldsymbol{h}_{t+1}\|_{\ell_2}^2] \leq \min\{n, p\} B_{t+1}^2.
$$

Finally, using the fact that $\boldsymbol{h}_t$ is $B_t$-Lipschitz function and utilizing Gaussian concentration of $\boldsymbol{q}_t \sim \mathcal{N}(0, \boldsymbol{I}_{tp})$, we find

$$
\mathbb{P}(\|\boldsymbol{h}_{t+1}\|_{\ell_2} - \mathbb{E}[\|\boldsymbol{h}_{t+1}\|_{\ell_2}] \geq t) \leq \exp(-\frac{t^2}{2B_t^2}).
$$

Setting $t = (c-1)\sqrt{m} B_t$ for sufficiently large $c > 0$, we find $\mathbb{P}(\|\boldsymbol{h}_t\|_{\ell_2} \geq \sqrt{n} B_t + (c-1)\sqrt{m} B_t) \leq \exp(-100m)$. $\qquad\square$

**Lemma B.4** (Odd activations). *Suppose $\phi$ is strictly increasing and obeys $\phi(x) = -\phi(-x)$ for all $x$ and $\boldsymbol{h}_0 = 0$. Consider the state equation (1.1) driven $\boldsymbol{u}_t \overset{\text{i.i.d.}}{\sim} \mathcal{N}(0, \boldsymbol{I}_p)$. We have that $\mathbb{E}[\boldsymbol{h}_t] = 0$.*

*Proof.* We will inductively show that $\{\boldsymbol{h}_t\}_{t \geq 0}$ has a symmetric distribution around 0. Suppose the vector $\boldsymbol{h}_t$ satisfies this assumption. Let $S \subset \mathbb{R}^n$ be a set. We will argue that $\mathbb{P}(\boldsymbol{h}_{t+1} \subset S) = \mathbb{P}(\boldsymbol{h}_{t+1} \subset -S)$. Since $\phi$ is strictly increasing, it is bijective on vectors, and we can define the unique inverse set $S' = \phi^{-1}(S)$. Also since $\phi$ is odd, $\phi(-S') = -S$. Since $\boldsymbol{h}_t, \boldsymbol{u}_t$ are independent and symmetric, we reach the desired conclusion as follows

$$
\mathbb{P}(\boldsymbol{h}_{t+1} \subset S) = \mathbb{P}(\boldsymbol{A}\boldsymbol{h}_t + \boldsymbol{B}\boldsymbol{u}_t \subset S') = \mathbb{P}(\boldsymbol{A}(-\boldsymbol{h}_t) + \boldsymbol{B}(-\boldsymbol{u}_t) \subset S')
\tag{B.5}
$$

$$
= \mathbb{P}(\boldsymbol{A}\boldsymbol{h}_t + \boldsymbol{B}\boldsymbol{u}_t \subset -S') = \mathbb{P}(\phi(\boldsymbol{A}\boldsymbol{h}_t + \boldsymbol{B}\boldsymbol{u}_t) \subset \phi(-S')) = \mathbb{P}(\boldsymbol{h}_{t+1} \subset -S).
\tag{B.6}
$$

$\qquad\square$

**Theorem B.5** (State-vector lower bound). *Consider the nonlinear state equation* (1.1) *with* $\{\boldsymbol{u}_t\}_{t \geq 0} \overset{i.i.d.}{\sim}$ $\mathcal{N}(0, \boldsymbol{I}_p)$. *Suppose* $\phi$ *is a* $\beta$-*increasing function for some constant* $\beta > 0$. *For any* $t \geq 1$, *the state vector obeys*

$$\boldsymbol{\Sigma}[\boldsymbol{h}_t] \succeq \beta^2 s_{\min}(\boldsymbol{B}\boldsymbol{B}^T)\boldsymbol{I}_n.$$

*Proof.* The proof is an application of Lemma B.7. The main idea is to write $\boldsymbol{h}_t$ as sum of two independent vectors, one of which has independent entries. Consider a multivariate Gaussian vector $\boldsymbol{g} \sim \mathcal{N}(0, \boldsymbol{\Sigma})$. $\boldsymbol{g}$ is statistically identical to $\boldsymbol{g}_1 + \boldsymbol{g}_2$ where $\boldsymbol{g}_1 \sim \mathcal{N}(0, s_{\min}(\boldsymbol{\Sigma})\boldsymbol{I}_d)$ and $\boldsymbol{g}_2 \sim \mathcal{N}(0, \boldsymbol{\Sigma} - s_{\min}(\boldsymbol{\Sigma})\boldsymbol{I}_d)$ are independent multivariate Gaussians.

Since $\boldsymbol{B}\boldsymbol{u}_t \sim \mathcal{N}(0, \boldsymbol{B}\boldsymbol{B}^T)$, setting $\boldsymbol{\Sigma} = \boldsymbol{B}\boldsymbol{B}^T$ and $s_{\min} = s_{\min}(\boldsymbol{\Sigma})$, we have that $\boldsymbol{B}\boldsymbol{u}_t \sim \boldsymbol{g}_1 + \boldsymbol{g}_2$ where $\boldsymbol{g}_1, \boldsymbol{g}_2$ are independent and $\boldsymbol{g}_1 \sim \mathcal{N}(0, s_{\min}\boldsymbol{I}_n)$ and $\boldsymbol{g}_2 \sim \mathcal{N}(0, \boldsymbol{\Sigma} - s_{\min}\boldsymbol{I}_n)$. Consequently, we may write

$$\boldsymbol{B}\boldsymbol{u}_t + \boldsymbol{A}\boldsymbol{h}_t \sim \boldsymbol{g}_1 + \boldsymbol{g}_2 + \boldsymbol{A}\boldsymbol{h}_t.$$

For lower bound, the crucial component will be the $\boldsymbol{g}_1$ term; which has i.i.d. entries. Applying Lemma B.7 by setting $\boldsymbol{x} = \boldsymbol{g}_1$ and $\boldsymbol{y} = \boldsymbol{g}_2 + \boldsymbol{A}\boldsymbol{h}_t$, and using the fact that $\boldsymbol{h}_t, \boldsymbol{g}_1, \boldsymbol{g}_2$ are all independent of each other, we find the advertised bound, for all $t \geq 0$, via

$$\boldsymbol{\Sigma}[\boldsymbol{h}_{t+1}] = \boldsymbol{\Sigma}[\phi(\boldsymbol{g}_1 + \boldsymbol{g}_2 + \boldsymbol{A}\boldsymbol{h}_t)] \succeq \beta^2 s_{\min}\boldsymbol{I}_n.$$

$\square$

The next theorem applies to multiple-input-single-output (MISO) systems where $\boldsymbol{A}$ is a scalar and $\boldsymbol{B}$ is a row vector. The goal is refining the lower bound of Theorem B.5.

**Theorem B.6** (MISO lower bound). *Consider the setup of Theorem B.5 with single output i.e.* $n = 1$. *For any* $t \geq 1$, *the state vector obeys*

$$\mathbf{var}[\boldsymbol{h}_t] \geq \beta^2 \|\boldsymbol{B}\|_{\ell_2}^2 \frac{1 - (\beta|\boldsymbol{A}|)^{2t}}{1 - \beta^2|\boldsymbol{A}|^2}.$$

*Proof.* For any random variable $X$, applying Lemma B.7, we have $\mathbf{var}[\phi(X)] \geq \beta^2\mathbf{var}[X]$. Recursively, this yields

$$\mathbf{var}[\boldsymbol{h}_{t+1}] = \mathbf{var}[\phi(\boldsymbol{A}\boldsymbol{h}_t + \boldsymbol{B}\boldsymbol{u}_t)] \geq \beta^2\mathbf{var}[\boldsymbol{A}\boldsymbol{h}_t + \boldsymbol{B}\boldsymbol{u}_t] = \beta^2(|\boldsymbol{A}|^2\mathbf{var}[\boldsymbol{h}_t] + \|\boldsymbol{B}\|_{\ell_2}^2).$$

Expanding these inequalities till $\boldsymbol{h}_0$, we obtain the desired bound

$$\mathbf{var}[\boldsymbol{h}_t] \geq \sum_{i=1}^{t}(\beta^i|\boldsymbol{A}|^{i-1}\|\boldsymbol{B}\|_{\ell_2})^2.$$

$\square$

**Lemma B.7** (Vector lower bound). *Suppose* $\phi$ *is a* $\beta$-*increasing function. Let* $\boldsymbol{x} = [\boldsymbol{x}_1 \ \ldots \ \boldsymbol{x}_n]^T$ *be a vector with i.i.d. entries distributed as* $\boldsymbol{x}_i \sim X$. *Let* $\boldsymbol{y}$ *be a random vector independent of* $\boldsymbol{x}$. *Then,*

$$\boldsymbol{\Sigma}[\phi(\boldsymbol{x} + \boldsymbol{y})] \succeq \beta^2\mathbf{var}[X]\boldsymbol{I}_n.$$

*Proof.* We first apply law of total covariance (e.g. Lemma B.8) to simplify the problem using the following lower bound based on the independence of $\boldsymbol{x}$ and $\boldsymbol{y}$,

$$\boldsymbol{\Sigma}[\phi(\boldsymbol{x} + \boldsymbol{y})] \succeq \mathbb{E}_{\boldsymbol{y}}[\boldsymbol{\Sigma}[\phi(\boldsymbol{x} + \boldsymbol{y}) \mid \boldsymbol{y}]] \tag{B.7}$$

$$= \mathbb{E}_{\boldsymbol{y}}[\boldsymbol{\Sigma}_{\boldsymbol{x}}[\phi(\boldsymbol{x} + \boldsymbol{y})]]. \tag{B.8}$$

Now, focusing on the covariance $\boldsymbol{\Sigma}_{\boldsymbol{x}}[\phi(\boldsymbol{x} + \boldsymbol{y})]$, fixing a realization of $\boldsymbol{y}$, and using the fact that $\boldsymbol{x}$ has i.i.d. entries; $\phi(\boldsymbol{x} + \boldsymbol{y})$ has independent entries as $\phi$ applies entry-wise. This implies that $\boldsymbol{\Sigma}_{\boldsymbol{x}}[\phi(\boldsymbol{x} + \boldsymbol{y})]$ is a diagonal matrix. Consequently, its lowest eigenvalue is the minimum variance over all entries,

$$\boldsymbol{\Sigma}_{\boldsymbol{x}}[\phi(\boldsymbol{x} + \boldsymbol{y})] \succeq \min_{1 \leq i \leq n} \mathbf{var}[\phi(\boldsymbol{x}_i + \boldsymbol{y}_i)]\boldsymbol{I}_n.$$

Fortunately, Lemma B.9 provides the lower bound $\mathbf{var}[\phi(\boldsymbol{x}_i + \boldsymbol{y}_i)] \geq \beta^2\mathbf{var}[X]$. Since this lower bound holds for any fixed realization of $\boldsymbol{y}$, it still holds after taking expectation over $\boldsymbol{y}$; which concludes the proof. $\square$

The next two lemmas are helper results for Lemma B.7 and are provided for the sake of completeness.

**Lemma B.8** (Law of total covariance). *Let* $\boldsymbol{x}, \boldsymbol{y}$ *be two random vectors and assume* $\boldsymbol{y}$ *has finite covariance. Then*

$$\boldsymbol{\Sigma}[\boldsymbol{y}] = \mathbb{E}[\boldsymbol{\Sigma}[\boldsymbol{y} \mid \boldsymbol{x}]] + \boldsymbol{\Sigma}[\mathbb{E}[\boldsymbol{y} \mid \boldsymbol{x}]].$$

*Proof.* First, write $\mathbf{\Sigma}[\boldsymbol{y}] = \mathbb{E}[\boldsymbol{y}\boldsymbol{y}^T] - \mathbb{E}[\boldsymbol{y}]\,\mathbb{E}[\boldsymbol{y}^T]$. Then, applying the law of total expectation to each term,

$$\mathbf{\Sigma}[\boldsymbol{y}] = \mathbb{E}[\mathbb{E}[\boldsymbol{y}\boldsymbol{y}^T \mid \boldsymbol{x}]] - \mathbb{E}[\mathbb{E}[\boldsymbol{y} \mid \boldsymbol{x}]]\,\mathbb{E}[\mathbb{E}[\boldsymbol{y}^T \mid \boldsymbol{x}]].$$

Next, we can write the conditional expectation as $\mathbb{E}[\mathbb{E}[\boldsymbol{y}\boldsymbol{y}^T \mid \boldsymbol{x}]] = \mathbb{E}[\mathbf{\Sigma}[\boldsymbol{y} \mid \boldsymbol{x}]] + \mathbb{E}[\mathbb{E}[\boldsymbol{y} \mid \boldsymbol{x}]\,\mathbb{E}[\boldsymbol{y} \mid \boldsymbol{x}]]^T$. To conclude, we obtain the covariance of $\mathbb{E}[\boldsymbol{y} \mid \boldsymbol{x}]$ via the difference,

$$\mathbb{E}[\mathbb{E}[\boldsymbol{y} \mid \boldsymbol{x}]\,\mathbb{E}[\boldsymbol{y} \mid \boldsymbol{x}]]^T - \mathbb{E}[\mathbb{E}[\boldsymbol{y} \mid \boldsymbol{x}]]\,\mathbb{E}[\mathbb{E}[\boldsymbol{y}^T \mid \boldsymbol{x}]] = \mathbf{\Sigma}[\mathbb{E}[\boldsymbol{y} \mid \boldsymbol{x}]],$$

which yields the desired bound. $\qquad\square$

**Lemma B.9** (Scalar lower bound). *Suppose $\phi$ is a $\beta$-increasing function with $\beta > 0$ as defined in Definition 3.1. Given a random variable $X$ and a scalar $y$, we have*

$$\mathbf{var}[\phi(X + y)] \geq \beta^2 \mathbf{var}[X].$$

*Proof.* Since $\phi$ is $\beta$-increasing, it is invertible and $\phi^{-1}$ is strictly increasing. Additionally, $\phi^{-1}$ is $1/\beta$ Lipschitz since,

$$|\phi(a) - \phi(b)| \geq \beta|a - b| \implies |a - b| \geq \beta|\phi^{-1}(a) - \phi^{-1}(b)|.$$

Using this observation and the fact that $\mathbb{E}[X]$ minimizes $\mathbb{E}(X - \alpha)^2$ over $\alpha$, $\mathbf{var}[\phi(X + y)]$ can be lower bounded as follows

$$\begin{aligned}
\mathbf{var}[\phi(X + y)] &= \mathbb{E}(\phi(X + y) - \mathbb{E}[\phi(X + y)])^2 \\
&\geq \beta^2 \, \mathbb{E}((X + y) - \phi^{-1}(\mathbb{E}[\phi(X + y)]))^2 \\
&\geq \beta^2 \, \mathbb{E}(X + y - \mathbb{E}[X + y])^2 \\
&= \beta^2 \, \mathbb{E}(X - \mathbb{E}\,X)^2 = \beta^2 \mathbf{var}[X].
\end{aligned}$$

Note that, the final line is the desired conclusion. $\qquad\square$

## C    TRUNCATING STABLE SYSTEMS

One of the challenges in analyzing dynamical systems is the fact that samples from the same trajectory have temporal dependence. This section shows that, for stable systems, the impact of the past states decay exponentially fast and the system can be approximated by using the recent inputs only. We first define the truncation of the state vector.

**Definition C.1** (Truncated state vector). *Suppose $\phi(0) = 0$, initial condition $h_0 = 0$, and consider the state equation (1.1). Given a timestamp $t$, $L$-truncation of the state vector $\boldsymbol{h}_t$ is denoted by $\bar{\boldsymbol{h}}_{t,L}$ and is equal to $\boldsymbol{q}_t$ where*

$$\boldsymbol{q}_{\tau+1} = \phi(\boldsymbol{A}\boldsymbol{q}_\tau + \boldsymbol{B}\boldsymbol{u}_\tau')  \quad, \quad q_0 = 0 \tag{C.1}$$

*is the state vector generated by the inputs $\boldsymbol{u}_\tau'$ satisfying*

$$\boldsymbol{u}_\tau' = \begin{cases} 0 \text{ if } \tau < t - L \\ \boldsymbol{u}_\tau \text{ else} \end{cases} .$$

In words, $L$ truncated state vector $\bar{\boldsymbol{h}}_{t,L}$ is obtained by unrolling $\boldsymbol{h}_t$ until time $t - L$ and setting the contribution of the state vector $\boldsymbol{h}_{t-L}$ to 0. This way, $\bar{\boldsymbol{h}}_{t,L}$ depends only on the variables $\{\boldsymbol{u}_\tau\}_{\tau=t-L}^{t-1}$.

The following lemma states that impact of truncation can be made fairly small for stable systems ($\|\boldsymbol{A}\| < 1$).

**Lemma C.2** (Truncation impact – deterministic). *Consider the state vector $\boldsymbol{h}_t$ and its $L$-truncation $\bar{\boldsymbol{h}}_{t,L}$ from Definition C.1. Suppose $\phi$ is 1-Lipschitz. We have that*

$$\|\boldsymbol{h}_t - \bar{\boldsymbol{h}}_{t,L}\|_{\ell_2} \leq \begin{cases} 0 \text{ if } t \leq L \\ \|\boldsymbol{A}\|^L \|\boldsymbol{h}_{t-L}\|_{\ell_2} \text{ else} \end{cases} .$$

*Proof.* When $t \leq L$, Definition C.1 implies $\boldsymbol{u}_\tau' = \boldsymbol{u}_\tau$ hence $\boldsymbol{h}_t = \boldsymbol{q}_t = \bar{\boldsymbol{h}}_{t,L}$. When $t > L$, we again use Definition C.1 and recall that $\boldsymbol{u}_\tau' = 0$ until time $\tau = t - L - 1$. For all $t - L < \tau \leq t$, using 1-Lipschitzness of $\phi$, we have that

$$\begin{aligned}
\|\boldsymbol{h}_\tau - \boldsymbol{q}_\tau\|_{\ell_2} &= \|\phi(\boldsymbol{A}\boldsymbol{h}_{\tau-1} + \boldsymbol{B}\boldsymbol{u}_{\tau-1}) - \phi(\boldsymbol{A}\boldsymbol{q}_{\tau-1} + \boldsymbol{B}\boldsymbol{u}_{\tau-1})\|_{\ell_2} \\
&\leq \|(\boldsymbol{A}\boldsymbol{h}_{\tau-1} + \boldsymbol{B}\boldsymbol{u}_{\tau-1}) - (\boldsymbol{A}\boldsymbol{q}_{\tau-1} + \boldsymbol{B}\boldsymbol{u}_{\tau-1})\|_{\ell_2} \\
&\leq \|\boldsymbol{A}(\boldsymbol{h}_{\tau-1} - \boldsymbol{q}_{\tau-1})\|_{\ell_2} \leq \|\boldsymbol{A}\|\|\boldsymbol{h}_{\tau-1} - \boldsymbol{q}_{\tau-1}\|_{\ell_2}.
\end{aligned}$$

Applying this recursion between $t - L < \tau \leq t$ and using the fact that $\boldsymbol{q}_{t-L} = 0$ implies the advertised result

$$\begin{aligned}
\|\boldsymbol{h}_t - \boldsymbol{q}_t\|_{\ell_2} &\leq \|\boldsymbol{A}\|^L \|\boldsymbol{h}_{t-L} - \boldsymbol{q}_{t-L}\|_{\ell_2} \\
&\leq \|\boldsymbol{A}\|^L \|\boldsymbol{h}_{t-L}\|_{\ell_2}.
\end{aligned}$$

$\qquad\square$

## C.1 NEAR INDEPENDENCE OF SUB-TRAJECTORIES

We will now argue that, for stable systems, a single trajectory can be split into multiple nearly independent trajectories. First, we describe how the sub-trajectories are constructed.

**Definition C.3** (Sub-trajectory). *Let sampling rate $L \geq 1$ and offset $1 \leq \bar{\tau} \leq L$ be two integers. Let $\bar{N} = \bar{N}_{\bar{\tau}}$ be the largest integer obeying $(\bar{N} - 1)L + \bar{\tau} \leq N$. We sample the trajectory $\{h_t, u_t\}_{t=0}^N$ at the points $\bar{\tau}, \bar{\tau} + L, \ldots, \bar{\tau} + (\bar{N} - 1)L + \bar{\tau}$ and define the $\bar{\tau}$th sub-trajectory as*

$$(h^{(i)}, u^{(i)}) := (h^{(i,\bar{\tau})}, u^{(i,\bar{\tau})}) = (h_{(i-1)L+\bar{\tau}}, u_{(i-1)L+\bar{\tau}}).$$

**Definition C.4** (Truncated sub-trajectory). *Consider the state equation (1.1) and recall Definition C.1. Given offset $\bar{\tau}$ and sampling rate $L$, for $1 \leq i \leq \bar{N}$, the $i$th truncated sub-trajectory states are $\{\bar{h}^{(i)}\}_{i=1}^{\bar{N}}$ where the $i$th state is defined as*

$$\bar{h}^{(i)} = \bar{h}_{L(i-1)+\bar{\tau}, L-1}.$$

The truncated samples are independent of each other as shown in the next lemma.

**Lemma C.5.** *Consider the truncated states of Definition C.4. If (1.1) is generated by independent vectors $\{u_t\}_{t\geq 0}$, for any offset $\bar{\tau}$ and sampling rate $L$, the vectors $\{\bar{h}^{(i)}\}_{i=1}^{\bar{N}}, \{u^{(i)}\}_{i=1}^{\bar{N}}$ are all independent of each other.*

*Proof.* By construction $\bar{h}^{(i)}$ only depends on the vectors $\{u_\tau\}_{\tau=L(i-2)+\bar{\tau}+1}^{L(i-1)+\bar{\tau}-1}$. Note that the dependence ranges $[L(i-2) + \bar{\tau} + 1, L(i-1) + \bar{\tau} - 1]$ are disjoint intervals for different $i$'s; hence $(\bar{h}^{(i)})_{i=1}^{\bar{N}}$ are independent of each other. To show the independence of $u^{(i)}$ and $\bar{h}^{(i)}$; observe that inputs $u^{(i)} = u_{L(i-1)+\bar{\tau}}$ have timestamp $\bar{\tau}$ modulo $L$; which is not covered by the dependence range of $(\bar{h}^{(i)})_{i=1}^{\bar{N}}$. $\square$

If the input is randomly generated, Lemma C.2 can be combined with a probabilistic bound on $h_t$, to show that truncated states $\bar{h}^{(i)}$ are fairly close to the actual states $h^{(i)}$.

**Lemma C.6** (Truncation impact – random). *Given offset $\bar{\tau}$ and sampling rate $L$, consider the state vectors of the sub-trajectory $\{h^{(i)}\}_{i=1}^{\bar{N}}$ and $L - 1$-truncations $(\bar{h}^{(i)})_{i=1}^{\bar{N}}$. Suppose $\{u_t\}_{t\geq 0} \overset{i.i.d.}{\sim} \mathcal{N}(0, I_p)$, $\|A\| < 1$, $h_0 = 0$, $\phi$ is 1-Lipschitz, and $\phi(0) = 0$. Also suppose upper bound (4.3) of Assumption 1 holds for some $\theta \leq \sqrt{n}, \gamma_+ > 0$. There exists an absolute constant $c > 0$ such that with probability at least $1 - 2\bar{N}\exp(-100n)$, for all $1 \leq i \leq \bar{N}$, the following bound holds*

$$\|h^{(i)} - \bar{h}^{(i)}\|_{\ell_2} \leq c\sqrt{n}\|A\|^{L-1}\sqrt{\gamma_+}.$$

*In particular, we can always pick $\gamma_+ = B_\infty^2$ (via Lemma B.3).*

*Proof.* Using Assumption 1, we can apply Lemma F.3 on vectors $\{h_{(i-2)L+\bar{\tau}+1}\}_{i=1}^{\bar{N}}$. Using a union bound, with desired probability, all vectors obey

$$\|h_{(i-2)L+\bar{\tau}+1} - \mathbb{E}[h_{(i-2)L+\bar{\tau}+1}]\|_{\ell_2} \leq (c-1)\sqrt{n\gamma_+},$$

for sufficiently large $c$. Since $\theta \leq \sqrt{n}$, triangle inequality implies $\|h_{(i-2)L+\bar{\tau}+1}\|_{\ell_2} \leq c\sqrt{n\gamma_+}$. Now, applying Lemma C.2, for all $1 \leq i \leq \bar{N}$, we find

$$\begin{aligned}
\|h^{(i)} - \bar{h}^{(i)}\|_{\ell_2} &= \|h_{(i-1)L+\bar{\tau}} - \bar{h}_{(i-1)L+\bar{\tau}, L-1}\|_{\ell_2} \\
&\leq \|A\|^{L-1}\|h_{(i-2)L+\bar{\tau}+1}\|_{\ell_2} \\
&\leq c\|A\|^{L-1}\sqrt{n\gamma_+}.
\end{aligned}$$

$\square$

# D PROPERTIES OF THE DATA MATRIX

This section utilizes the probabilistic estimates from Section B to provide bounds on the condition number of data matrices obtained from the RNN trajectory (1.1). Following (2.2), these matrices $H, U$ and $X$ are defined as

$$H = [h_1 \ \ldots \ h_N]^T \quad , \quad U = H = [u_1 \ \ldots \ u_N]^T \quad , \quad X = [x_1 \ \ldots \ x_N]^T. \tag{D.1}$$

The challenge is that, the state matrix $H$ has dependent rows; which will be addressed by carefully splitting the trajectory $\{u_t, h_t\}_{t=0}^N$ into multiple sub-trajectories which are internally weakly dependent as discussed in Section C. We first define the matrices obtained from these sub-trajectories.

**Definition D.1.** *Given sampling rate $L$ and offset $\bar{\tau}$, consider the $L$-subsampled trajectory $\{\boldsymbol{h}^{(i)}, \boldsymbol{u}^{(i)}\}_{i=1}^{\bar{N}}$ as described in Definitions C.3 and C.4. Define the matrices $\bar{\boldsymbol{H}} = \bar{\boldsymbol{H}}^{(\bar{\tau})} \in \mathbb{R}^{\bar{N} \times n}$, $\tilde{\boldsymbol{H}} = \tilde{\boldsymbol{H}}^{(\bar{\tau})} \in \mathbb{R}^{\bar{N} \times n}$, $\tilde{\boldsymbol{U}} = \tilde{\boldsymbol{U}}^{(\bar{\tau})} \in \mathbb{R}^{\bar{N} \times p}$, and $\tilde{\boldsymbol{X}} = \tilde{\boldsymbol{X}}^{(\bar{\tau})} \in \mathbb{R}^{\bar{N} \times (n+p)}$ as*

$$\bar{\boldsymbol{H}} = [\bar{\boldsymbol{h}}^{(1)} \ \ldots \ \bar{\boldsymbol{h}}^{(\bar{N})}]^T, \ \tilde{\boldsymbol{H}} = [\boldsymbol{h}^{(1)} \ \ldots \ \boldsymbol{h}^{(\bar{N})}]^T, \ \tilde{\boldsymbol{U}} = [\boldsymbol{u}^{(1)} \ \ldots \ \boldsymbol{u}^{(\bar{N})}]^T, \ \tilde{\boldsymbol{X}} = [\mu \tilde{\boldsymbol{H}} \ \tilde{\boldsymbol{U}}].$$

**Lemma D.2** (Handling perturbation). *Consider the nonlinear state equation* (1.1). *Given sampling rate $L > 0$ and offset $\bar{\tau}$, consider the matrices $\bar{\boldsymbol{H}}, \tilde{\boldsymbol{H}}, \tilde{\boldsymbol{X}}$ of Definition D.1 and let $\boldsymbol{Q} = [\gamma_+^{-1/2} \bar{\boldsymbol{H}} \ \tilde{\boldsymbol{U}}] \in \mathbb{R}^{\bar{N} \times (n+p)}$. Suppose Assumption 1 holds, $\phi$ is $\beta$-increasing, and $\boldsymbol{u}_t \overset{i.i.d.}{\sim} \mathcal{N}(0, \boldsymbol{I}_p)$. There exists an absolute constant $C > 0$ such that if $\bar{N} \geq C \frac{\gamma_+^2}{\gamma_-^2}(n + p)$, with probability $1 - 8\exp(-c\frac{\gamma_-^2}{\gamma_+^2}\bar{N})$, for all matrices $\boldsymbol{M}$ obeying $\|\boldsymbol{M} - \bar{\boldsymbol{H}}\| \leq \frac{\sqrt{\gamma_- \bar{N}}}{10}$, the perturbed $\boldsymbol{Q}$ matrices given by,*

$$\tilde{\boldsymbol{Q}} = [\gamma_+^{-1/2} \boldsymbol{M} \ \tilde{\boldsymbol{U}}], \tag{D.2}$$

*satisfy*

$$(\Theta + \sqrt{2})^2 \succeq \frac{\tilde{\boldsymbol{Q}}^T \tilde{\boldsymbol{Q}}}{\bar{N}} \succeq \frac{\gamma_-}{2\gamma_+}. \tag{D.3}$$

*Proof.* This result is a direct application of Theorem F.1 after determining minimum/maximum eigenvalues of population covariance. The cross covariance obeys $\mathbb{E}[\bar{\boldsymbol{H}}^T \tilde{\boldsymbol{U}}] = 0$ due to independence. Also, for $i > 1$, the truncated state vector $\bar{\boldsymbol{h}}^{(i)}$ is statistically identical to $\boldsymbol{h}_{L-1}$ hence $\boldsymbol{\Sigma}[\bar{\boldsymbol{h}}^{(i)}] \succeq \gamma_- \boldsymbol{I}_n$. Consequently, $\boldsymbol{\Sigma}[\boldsymbol{u}^{(i)}] = \boldsymbol{I}_p$, $\frac{1}{\gamma_+}\boldsymbol{\Sigma}[\bar{\boldsymbol{h}}^{(i)}] \preceq \boldsymbol{I}_n$ for all $i$ and $\frac{\gamma_-}{\gamma_+}\boldsymbol{I}_n \preceq \frac{1}{\gamma_+}\boldsymbol{\Sigma}[\bar{\boldsymbol{h}}^{(i)}]$ for all $i > 1$. Hence, setting $\boldsymbol{q}_i = \begin{bmatrix} \frac{1}{\sqrt{\gamma_+}}\bar{\boldsymbol{h}}^{(i)} \\ \boldsymbol{u}^{(i)} \end{bmatrix}$, for all $i > 1$

$$\frac{\gamma_-}{\gamma_+}\boldsymbol{I}_n \preceq \boldsymbol{\Sigma}[\boldsymbol{q}_i] \preceq \boldsymbol{I}_n.$$

Set the matrix $\bar{\boldsymbol{Q}} = [\boldsymbol{q}_2 \ \ldots \ \boldsymbol{q}_{\bar{N}}]^T$ and note that $\boldsymbol{Q} = [\boldsymbol{q}_1 \ \bar{\boldsymbol{Q}}^T]^T$. Applying Theorem F.1 on $\bar{\boldsymbol{Q}}$ and Corollary F.2 on $\boldsymbol{Q}$, we find that, with the desired probability,

$$\theta + \sqrt{3/2} \geq \frac{1}{\sqrt{\bar{N}}}\|\boldsymbol{Q}\| \geq \frac{1}{\sqrt{\bar{N}}}s_{\min}(\boldsymbol{Q}) \geq \frac{1}{\sqrt{\bar{N}}}s_{\min}(\bar{\boldsymbol{Q}}) \geq \sqrt{\frac{N-1}{N}}\sqrt{\frac{2\gamma_-}{3\gamma_+}} \geq 0.99 \times \sqrt{\frac{2\gamma_-}{3\gamma_+}}.$$

Setting $\boldsymbol{E} = \boldsymbol{M} - \bar{\boldsymbol{H}}$ and observing $\tilde{\boldsymbol{Q}} = \boldsymbol{Q} + [\gamma_+^{-1/2}\boldsymbol{E} \ 0]$, the impact of the perturbation $\boldsymbol{E}$ can be bounded naively via $s_{\min}(\boldsymbol{Q}) - \gamma_+^{-1/2}\|\boldsymbol{E}\| \leq s_{\min}(\tilde{\boldsymbol{Q}}) \leq \|\tilde{\boldsymbol{Q}}\| \leq \|\boldsymbol{Q}\| + \gamma_+^{-1/2}\|\boldsymbol{E}\|$. Using the assumed bound on $\|\boldsymbol{E}\|$, this yields

$$\theta + \sqrt{2} \geq \frac{1}{\sqrt{\bar{N}}}\|\tilde{\boldsymbol{Q}}\| \geq \frac{1}{\sqrt{\bar{N}}}s_{\min}(\tilde{\boldsymbol{Q}}) \geq \sqrt{\frac{\gamma_-}{2\gamma_+}}.$$

This final inequality is identical to the desired bound (D.3). $\qquad\square$

**Theorem D.3** (Data matrix condition). *Consider the nonlinear state-equation* (1.1). *Given $\gamma_+ \geq \gamma_- > 0$, define the condition number $\rho = \frac{\gamma_+}{\gamma_-}$. For some absolute constants $c, C > 0$, pick a trajectory length $N$ where*

$$L = \lceil 1 - \frac{\log(cn\rho)}{\log\|\boldsymbol{A}\|}\rceil \quad, \quad N_0 = \lfloor\frac{N}{L}\rfloor \geq C\rho^2(n+p),$$

*and pick scaling $\mu = \frac{1}{\sqrt{\gamma_+}}$. Suppose $\|\boldsymbol{A}\| < 1$, $\phi$ is $\beta$-increasing, $\boldsymbol{u}_t \overset{i.i.d.}{\sim} \mathcal{N}(0, \boldsymbol{I}_p)$, and Assumption 1 holds with $\gamma_+, \gamma_-, \theta, L$. Matrix $\boldsymbol{X} = [\boldsymbol{x}_1 \ \ldots \ \boldsymbol{x}_N]^T$ of (D.1) satisfies the following with probability $1 - 4N\exp(-100n) - 8L\exp(-\mathcal{O}(N_0/\rho^2))$.*

- *Each row of $\boldsymbol{X}$ has $\ell_2$ norm at most $c_0\sqrt{p+n}$ where $c_0$ is an absolute constant.*

- *$\boldsymbol{X}^T\boldsymbol{X}$ obeys the bound*

$$(\Theta + \sqrt{2})^2\boldsymbol{I}_{n+p} \succeq \frac{\boldsymbol{X}^T\boldsymbol{X}}{N} \succeq \rho^{-1}\boldsymbol{I}_{n+p}/2. \tag{D.4}$$

*Proof.* The first statement on $\ell_2$-norm bound can be concluded from Lemma D.4 and holds with probability $1 - 2N\exp(-100(n+p))$. To show the second statement, for a fixed offset $1 \leq \bar{\tau} \leq L$, consider Definition D.1 and the matrices $\tilde{\boldsymbol{H}}^{(\bar{\tau})}, \tilde{\boldsymbol{U}}^{(\bar{\tau})}, \tilde{\boldsymbol{X}}^{(\bar{\tau})}$. Observe that $\boldsymbol{X}$ is obtained by merging multiple sub-trajectory matrices

$\{\tilde{\boldsymbol{X}}^{(\bar{\tau})}\}_{\bar{\tau}=1}^{L}$. We will first show the advertised bound for an individual $\tilde{\boldsymbol{X}}^{(\bar{\tau})}$ by applying Lemma D.2 and then apply Lemma A.1 to obtain the bound on the combined matrix $\boldsymbol{X}$.

Recall that $\bar{N}_{\bar{\tau}}$ is the length of the $\bar{\tau}$th sub-trajectory i.e. number of rows of $\tilde{\boldsymbol{X}}^{(\bar{\tau})}$. By construction $2N_0 \geq \bar{N}_{\bar{\tau}} \geq N_0$ for all $1 \leq \bar{\tau} \leq L$. Given $1 \leq \bar{\tau} \leq L$ and triple $\bar{\boldsymbol{H}}^{(\bar{\tau})}, \tilde{\boldsymbol{H}}^{(\bar{\tau})}, \tilde{\boldsymbol{U}}^{(\bar{\tau})}$, set $\boldsymbol{Q} = [\mu \bar{\boldsymbol{H}}^{(\bar{\tau})} \; \tilde{\boldsymbol{U}}^{(\bar{\tau})}]$. Since $N_0$ is chosen to be large enough, applying Theorem D.2 with $\mu = 1/\sqrt{\gamma_+}$ choice, and noting $\rho = \gamma_+/\gamma_-$, we find that, with probability $1 - 4\exp(-c_1 N_0/\rho^2)$, all matrices $\boldsymbol{M}$ satisfying $\|\boldsymbol{M} - \bar{\boldsymbol{H}}^{(\bar{\tau})}\| \leq \sqrt{\gamma_- N_0}/10$ and $\tilde{\boldsymbol{Q}}$ as in (D.2) obeys

$$(\Theta + \sqrt{2})^2 \succeq \frac{\tilde{\boldsymbol{Q}}^T \tilde{\boldsymbol{Q}}}{N} \succeq \rho^{-1}/2. \tag{D.5}$$

Let us call this Event 1. To proceed, we will argue that with high probability $\|\tilde{\boldsymbol{H}}^{(\bar{\tau})} - \bar{\boldsymbol{H}}^{(\bar{\tau})}\|$ is small so that the bound above is applicable with $\boldsymbol{M} = \tilde{\boldsymbol{H}}^{(\bar{\tau})}$ choice; which sets $\tilde{\boldsymbol{Q}} = \tilde{\boldsymbol{X}}^{(\bar{\tau})}$ in (D.5). Applying Lemma C.6, we find that, with probability $1 - 2\bar{N}_{\bar{\tau}} \exp(-100n)$,

$$\|\bar{\boldsymbol{H}}^{(\bar{\tau})} - \tilde{\boldsymbol{H}}^{(\bar{\tau})}\| \leq \sqrt{2N_0} \max\{\|\boldsymbol{h}^{(i)} - \bar{\boldsymbol{h}}^{(i)}\|_{\ell_2}\} \leq c_0 \sqrt{2N_0}\sqrt{n\gamma_+}\|\boldsymbol{A}\|^{L-1}.$$

Let us call this Event 2. We will show that our choice of $L$ ensures right hand side is small enough and guarantees $\|\bar{\boldsymbol{H}}^{(\bar{\tau})} - \tilde{\boldsymbol{H}}^{(\bar{\tau})}\| \leq \sqrt{\gamma_- N_0}/10$. Set $c = \max\{200c_0^2, 1\}$. Desired claim follows by taking logarithms of upper/lower bounds and cancelling out $\sqrt{N_0}$ terms as follows

$$c_0\sqrt{n}\|\boldsymbol{A}\|^{L-1}\sqrt{\gamma_+} \leq \sqrt{\gamma_-}/10\sqrt{2} \iff (L-1)\log\|\boldsymbol{A}\| + \log\sqrt{cn\rho} \leq 0 \tag{D.6}$$

$$\iff -\frac{\log cn\rho}{2\log\|\boldsymbol{A}\|} \leq L - 1 \tag{D.7}$$

$$\impliedby L = \lceil 1 - \frac{\log(cn\rho)}{\log\|\boldsymbol{A}\|}\rceil. \tag{D.8}$$

Here we use the fact that $\log\|\boldsymbol{A}\| < 0$ since $\|\boldsymbol{A}\| < 1$ and $cn\rho \geq 0$. Consequently, both Event 1 and Event 2 hold with probability $1 - 4\exp(-c_1 N_0/\rho^2) - 2\bar{N}_{\bar{\tau}}\exp(-100n)$, implying (D.5) holds with $\tilde{\boldsymbol{Q}} = \tilde{\boldsymbol{X}}^{(\bar{\tau})}$. Union bounding this over $1 \leq \bar{\tau} \leq L$, (D.5) uniformly holds with $\tilde{\boldsymbol{Q}} = \tilde{\boldsymbol{X}}^{(\bar{\tau})}$ and all rows of $\boldsymbol{X}$ are $\ell_2$-bounded with probability $1 - 4N\exp(-100n) - 8L\exp(-c_1 N_0/\rho^2)$. Applying Lemma A.1 on $(\tilde{\boldsymbol{X}}^{(\bar{\tau})})_{\bar{\tau}=1}^{L}$, we conclude with the bound (D.4) on the merged matrix $\boldsymbol{X}$. □

**Lemma D.4** ($\ell_2$-bound on rows). *Consider the setup of Theorem D.3. With probability $1 - 2N\exp(-100(n + p))$, each row of $\boldsymbol{X}$ has $\ell_2$-norm at most $c\sqrt{p + n}$ for some constant $c > 0$.*

*Proof.* The $t$th row of $\boldsymbol{X}$ is equal to $\boldsymbol{x}_t = [\frac{\boldsymbol{h}_t^T}{\sqrt{\gamma_+}} \; \boldsymbol{u}_t^T]^T$. Since $\|\boldsymbol{h}_t - \mathbb{E}[\boldsymbol{h}_t]\|_{\psi_2} \leq \mathcal{O}(\sqrt{\gamma_+})$ and $\|\boldsymbol{u}_t\|_{\psi_2} \leq \mathcal{O}(1)$, we have that $\|\boldsymbol{x}_t - \mathbb{E}[\boldsymbol{x}_t]\|_{\psi_2} \leq \mathcal{O}(1)$. Now, applying Lemma F.3 on all rows $\{\boldsymbol{x}_t\}_{t=1}^{N}$, and using a union bound, with probability at least $1 - 2N\exp(-100(n + p))$, we have that $\|\boldsymbol{x}_t - \mathbb{E}[\boldsymbol{x}_t]\|_{\ell_2} \leq c\sqrt{n + p}$ for all $t$. To conclude, note that $\|\mathbb{E}[\boldsymbol{x}_t]\|_{\ell_2} = \|\mathbb{E}[\boldsymbol{h}_t]\|_{\ell_2}/\sqrt{\gamma_+} \leq \theta \leq 3\sqrt{n}$ via Assumption 1. □

# E   PROOFS OF MAIN RESULTS

## E.1   PROOF OF LEMMA 3.2

The statement follows from upper bound Lemma B.3 and lower bound Lemma B.5.

## E.2   PROOF OF THEOREM 4.2

*Proof.* To prove this theorem, we combine Theorem D.3 with deterministic SGD convergence result of Theorem 4.1. Applying Theorem D.3, with the desired probability, inequality (D.4) holds and for all $i$, input data satisfies the bound $\|\boldsymbol{x}_i\|_{\ell_2} \leq \sqrt{(n + p)/(2c_0)}$ for a sufficiently small constant $c_0 > 0$. As the next step, we will argue that these two events imply the convergence of SGD.

Let $\boldsymbol{\theta}^{(i)}, \boldsymbol{c}^{(i)} \in \mathbb{R}^{n+p}$ denote the $i$th rows of $\boldsymbol{\Theta}, \boldsymbol{C}$ respectively. Observe that the square-loss is separable along the rows of $\boldsymbol{C}$ via $\|\boldsymbol{\Theta} - \boldsymbol{C}\|_F^2 = \sum_{i=1}^{n} \|\boldsymbol{\theta}^{(i)} - \boldsymbol{c}^{(i)}\|_{\ell_2}^2$. Hence, SGD updates each row $\boldsymbol{c}^{(i)}$ via its own state equation

$$\boldsymbol{y}_{t,i} = \phi(\langle \boldsymbol{c}^{(i)}, \boldsymbol{x}_t \rangle),$$

where $\boldsymbol{y}_{t,i}$ is the $i$th entry of $\boldsymbol{y}_t$. Consequently, we can establish the convergence result for an individual row of $\boldsymbol{C}$. Convergence of all individual rows will imply the convergence of the overall matrix $\boldsymbol{\Theta}_\tau$ to the ground

truth $C$. Pick a row index $i$ ($1 \leq i \leq n$), set $c = c^{(i)}$ and denote $i$th row of $\Theta_\tau$ by $\theta_\tau$. Also denote the label corresponding to $i$th row by $y_t = y_{t,i}$. With this notation, SGD over (2.3) runs SGD over the $i$th row with equations $y_t = \phi(\langle c, x_t \rangle)$ and with loss functions

$$\mathcal{L}(\boldsymbol{\theta}) = N^{-1} \sum_{t=1}^N \mathcal{L}_t(\boldsymbol{\theta}), \ \mathcal{L}_t(\boldsymbol{\theta}) = \frac{1}{2}(y_t - \phi(\langle \boldsymbol{\theta}, x_t \rangle))^2.$$

Substituting our high-probability bounds on $x_t$ (e.g. (D.4)) into Theorem 4.1, we can set $B = (n+p)/(2c_0)$, $\gamma_+ = (\theta + \sqrt{2})^2$, and $\gamma_- = \rho^{-1}/2$. Consequently, using the learning rate $\eta = c_0 \frac{\beta^2 \rho^{-1}}{(\theta + \sqrt{2})^2 (n+p)}$, for all $\tau \geq 0$, the $\tau$th SGD iteration $\boldsymbol{\theta}_\tau$ obeys

$$\mathbb{E}[\|\boldsymbol{\theta}_\tau - c\|_{\ell_2}^2] \leq \|\boldsymbol{\theta}_0 - c\|_{\ell_2}^2 (1 - c_0 \frac{\beta^4 \rho^{-2}}{2(\theta + \sqrt{2})^2 (n+p)})^\tau, \tag{E.1}$$

where the expectation is over the random selection of SGD updates. This establishes the convergence for a particular row of $C$. Summing up these inequalities (E.1) over all rows $\boldsymbol{\theta}_\tau^{(1)}, \ldots, \boldsymbol{\theta}_\tau^{(n)}$ (which converge to $c^{(1)}, \ldots, c^{(n)}$ respectively) yields the targeted bound (4.4). $\qquad\square$

### E.3 PROOFS OF MAIN RESULTS ON STABLE SYSTEMS

#### E.3.1 PROOF OF THEOREM 3.3

*Proof.* Applying Lemmas B.3 and 3.2, independent of $L$, Assumption 1 holds with parameters
$$\gamma_+ = B_\infty^2 \quad , \quad \gamma_- = \beta^2 s_{\min}(\boldsymbol{B})^2 \quad , \quad \theta = \sqrt{6n} - \sqrt{2} \geq \sqrt{n}.$$
This yields $(\theta + \sqrt{2})^2 = 6n$. Hence, we can apply Theorem 4.2 with the learning rate $\eta = c_0 \frac{\beta^2}{6\rho n(n+p)}$ where

$$\rho = \frac{B_\infty^2}{\beta^2 s_{\min}(\boldsymbol{B})^2} = \frac{\gamma_+}{\gamma_-}, \tag{E.2}$$

and convergence rate $1 - \frac{\beta^2 \eta}{2\rho}$. To conclude with the stated result, we use the change of variable $c_0/6 \to c_0$. $\quad\square$

#### E.3.2 PROOF OF THEOREM 3.4

*Proof.* The proof is similar to that of Theorem 3.3. Applying Lemmas B.3, B.4, and 3.2, independent of $L$, Assumption 1 holds with parameters
$$\gamma_+ = B_\infty^2 \quad , \quad \gamma_- = s_{\min}(\boldsymbol{B})^2 \quad , \quad \theta = 0.$$
Hence, we again apply Theorem 4.2 with the learning rate $\eta = c_0 \frac{\beta^2}{2\rho(n+p)}$ where $\rho$ is given by (E.2). Use the change of variable $c_0/2 \to c_0$ to conclude with the stated result. $\qquad\square$

### E.4 LEARNING UNSTABLE SYSTEMS

In a similar fashion to Section 4, we provide a more general result on unstable systems that makes a parametric assumption on the statistical properties of the state vector.

**Assumption 2** (Well-behaved state vector – single timestamp). *Given timestamp $T_0 > 0$, there exists positive scalars $\gamma_+, \gamma_-, \theta$ and an absolute constant $C > 0$ such that $\theta \leq 3\sqrt{n}$ and the following holds*
$$\gamma_+ \boldsymbol{I}_n \succeq \boldsymbol{\Sigma}[h_{T_0}] \succeq \gamma_- \boldsymbol{I}_n \quad , \quad \|h_{T_0} - \mathbb{E}[h_{T_0}]\|_{\psi_2} \leq C\sqrt{\gamma_+} \quad and \quad \|\mathbb{E}[h_t]\|_{\ell_2} \leq \theta \sqrt{\gamma_+}. \tag{E.3}$$

The next theorem provides the parametrized result on unstable systems based on this assumption.

**Theorem E.1** (Unstable system - general). *Suppose we are given $N$ independent trajectories $(h_t^{(i)}, u_t^{(i)})_{t \geq 0}$ for $1 \leq i \leq N$. Sample each trajectory at time $T_0$ to obtain $N$ samples $(y_i, h_i, u_i)_{i=1}^N$ where $i$th sample is*
$$(y_i, h_i, u_i) = (h_{T_0+1}^{(i)}, h_{T_0}^{(i)}, u_{T_0}^{(i)}).$$
*Let $C, c_0 > 0$ be absolute constants. Suppose Assumption 1 holds with $T_0$ and sample size satisfies $N \geq C\rho^2(n+p)$ where $\rho = \gamma_+/\gamma_-$. Assume $\phi$ is $\beta$-increasing, zero initial state conditions, and $u_t \overset{i.i.d.}{\sim} \mathcal{N}(0, \boldsymbol{I}_p)$. Set scaling to be $\mu = 1/\sqrt{\gamma_+}$ and learning rate to be $\eta = c_0 \frac{\beta^2}{\rho(\theta+\sqrt{2})^2(n+p)}$. Starting from $\Theta_0$, we run SGD over the equations described in (2.2) and (2.3). With probability $1 - 2N \exp(-100(n+p)) - 4\exp(-\mathcal{O}(\frac{N}{\rho^2}))$, all iterates satisfy*

$$\mathbb{E}[\|\Theta_i - \boldsymbol{C}\|_F^2] \leq (1 - c_0 \frac{\beta^4}{2\rho^2(\theta+\sqrt{2})^2(n+p)})^\tau \|\Theta_0 - \boldsymbol{C}\|_F^2,$$

*where the expectation is over the randomness of the SGD updates.*

*Proof.* Set $\boldsymbol{x}_i = [\gamma_+^{-1/2}\boldsymbol{h}_i^T \; \boldsymbol{u}_i^T]^T$ and $\boldsymbol{X} = [\boldsymbol{x}_1 \; \dots \; \boldsymbol{x}_N]^T$. Since $\boldsymbol{X}$ has i.i.d. rows, we can apply Theorem F.1 and Lemma F.3 to find with the desired probability that

- Rows of $\boldsymbol{x}_i$ satisfy $\|\boldsymbol{x}_i - \mathbb{E}[\boldsymbol{x}_i]\|_{\psi_2} \leq \mathcal{O}(1)$ and $\mathbb{E}[\|\boldsymbol{x}_i\|_{\ell_2}] \leq 3\sqrt{n}$, hence all rows of $\boldsymbol{X}$ obeys $\|\boldsymbol{x}_i\|_{\ell_2} \leq \sqrt{(n+p)/(2c_0)}$,

- $\boldsymbol{X}$ satisfies

$$(\theta + \sqrt{2})^2 \succeq \frac{\boldsymbol{X}^T\boldsymbol{X}}{N} \succeq \rho^{-1}/2.$$

To proceed, using $\gamma_- = \rho^{-1}/2$, $B = (n+p)/(2c_0)$, and $\gamma_+ = (\theta + \sqrt{2})^2$, we apply Theorem 4.1 on the loss function (2.3); which yields the desired result. □

### E.5  PROOF OF THEOREM 5.1

*Proof.* The proof is a corollary of Theorem E.1. We need to substitute the proper values in Assumption 2. Applying Lemma B.3, we can substitute $\gamma_+ = B_{T_0}^2$ and $\theta = \sqrt{6n} - \sqrt{2} \geq \sqrt{n}$. Next, we need to find a lower bound. Applying Lemma 3.2 for $n > 1$ and Lemma B.6 for $n = 1$, we can substitute $\gamma_- = \gamma_+/\rho$ with the $\rho$ definition of (5.2). With these, the result follows as an immediate corollary of Theorem E.1. □

## F  SUPPLEMENTARY STATISTICAL RESULTS

The following theorem bounds the empirical covariance of matrices with independent subgaussian rows. Given a random vector $\boldsymbol{x}$, define the de-biasing operation as $\mathbf{zm}(\boldsymbol{x}) = \boldsymbol{x} - \mathbb{E}[\boldsymbol{x}]$.

**Theorem F.1.** *Let $\boldsymbol{A} \in \mathbb{R}^{n \times d}$ be a matrix with independent subgaussian rows $\{\boldsymbol{a}_i\}_{i=1}^n$ satisfying $\|\mathbf{zm}(\boldsymbol{a}_i)\|_{\psi_2} \leq \mathcal{O}(K)$ and $\boldsymbol{\Sigma}[\boldsymbol{a}_i] \preceq K^2\boldsymbol{I}_d$ for some $K > 0$ and $\|\mathbb{E}[\boldsymbol{a}_i]\|_{\ell_2} \leq \theta$. Suppose $\boldsymbol{\Sigma}[\boldsymbol{a}_i] \succeq \lambda\boldsymbol{I}_d$. Suppose $n \geq \mathcal{O}(K^4 d/\lambda^2)$. Then, each of the following happens with probability at least $1 - 2\exp(-cK^{-4}\lambda^2 n)$,*

- $\theta + \sqrt{3/2}K \geq \frac{1}{\sqrt{n}}\|\boldsymbol{A}\|$.

- *Suppose all rows of $\boldsymbol{A}$ have equal expectations. Then $\frac{1}{\sqrt{n}}s_{\min}(\boldsymbol{A}) \geq \sqrt{2\lambda/3}$.*

*Proof.* Let $\boldsymbol{E} = \mathbb{E}[\boldsymbol{A}]$, $\bar{\boldsymbol{A}} = \boldsymbol{A} - \mathbb{E}[\boldsymbol{A}]$, $\bar{\boldsymbol{a}}_i = \mathbf{zm}(\boldsymbol{a}_i)$. We will decompose $\boldsymbol{A} = \bar{\boldsymbol{A}} + \boldsymbol{E}$ hence we will first focus on bounding the upper and lower singular values of $\bar{\boldsymbol{A}}$ by studying the random processes $X_{\boldsymbol{v}} = \|\bar{\boldsymbol{A}}\boldsymbol{v}\|_{\ell_2}^2$ and $Y_{\boldsymbol{v}} = X_{\boldsymbol{v}} - \mathbb{E}[X_{\boldsymbol{v}}]$ over the unit sphere $\mathcal{S}^{d-1}$. First, we provide a deviation bound for the quantity $\sup_{\boldsymbol{v} \in \mathcal{S}^{d-1}} |Y_{\boldsymbol{v}}|$. To achieve this, we will utilize Talagrand's mixed tail bound and show that increments of $Y_{\boldsymbol{v}}$ are subexpoential. Pick two unit vectors $\boldsymbol{v}, \boldsymbol{u} \in \mathbb{R}^d$. Write $\boldsymbol{x} = \boldsymbol{u} + \boldsymbol{v}, \boldsymbol{y} = \boldsymbol{u} - \boldsymbol{v}$. We have that

$$X_{\boldsymbol{u}} - X_{\boldsymbol{v}} = \|\bar{\boldsymbol{A}}\boldsymbol{u}\|_{\ell_2}^2 - \|\bar{\boldsymbol{A}}\boldsymbol{v}\|_{\ell_2}^2 = \|\bar{\boldsymbol{A}}(\boldsymbol{x}+\boldsymbol{y})/2\|_{\ell_2}^2 - \|\bar{\boldsymbol{A}}(\boldsymbol{x}-\boldsymbol{y})/2\|_{\ell_2}^2 = \boldsymbol{x}^T\bar{\boldsymbol{A}}^T\bar{\boldsymbol{A}}\boldsymbol{y} = \sum_{i=1}^n (\bar{\boldsymbol{a}}_i^T\boldsymbol{x})(\bar{\boldsymbol{a}}_i^T\boldsymbol{y}).$$

Letting $\hat{\boldsymbol{x}} = \boldsymbol{x}/\|\boldsymbol{x}\|_{\ell_2}, \hat{\boldsymbol{y}} = \boldsymbol{y}/\|\boldsymbol{y}\|_{\ell_2}$, observe that, multiplication of subgaussians $\boldsymbol{x}^T\bar{\boldsymbol{a}}_i, \boldsymbol{y}^T\bar{\boldsymbol{a}}_i$ obey

$$\|(\boldsymbol{x}^T\bar{\boldsymbol{a}}_i)(\boldsymbol{y}^T\bar{\boldsymbol{a}}_i)\|_{\psi_1} \leq \mathcal{O}(\|\boldsymbol{x}\|_{\ell_2}\|\boldsymbol{y}\|_{\ell_2}K^2) \leq \mathcal{O}(K^2\|\boldsymbol{y}\|_{\ell_2}).$$

Centering this subexponential variable around zero introduces a factor of 2 when bounding subexponential norm and yields $\|(\boldsymbol{x}^T\bar{\boldsymbol{a}}_i)(\boldsymbol{y}^T\bar{\boldsymbol{a}}_i) - \mathbb{E}[(\boldsymbol{x}^T\bar{\boldsymbol{a}}_i)(\boldsymbol{y}^T\bar{\boldsymbol{a}}_i)]\|_{\psi_1} \leq \mathcal{O}(K^2\|\boldsymbol{y}\|_{\ell_2})$. Now, using the fact that $Y_{\boldsymbol{u}} - Y_{\boldsymbol{v}}$ is sum of $n$ independent zero-mean subexponential random variables, we have the tail bound

$$\mathbb{P}(n^{-1}|Y_{\boldsymbol{u}} - Y_{\boldsymbol{v}}| \geq t) \leq 2\exp(-c'n\min\{\frac{t^2}{K^4\|\boldsymbol{y}\|_{\ell_2}^2}, \frac{t}{K^2\|\boldsymbol{y}\|_{\ell_2}}\}).$$

Applying Talagrand's chaining bound for mixed tail processes with distance metrics $\rho_2 = \frac{K^2\|\cdot\|_{\ell_2}}{\sqrt{n}}, \rho_1 = \frac{K^2\|\cdot\|_{\ell_2}}{n}$, (Theorem 3.5 of Dirksen (2013) or Theorem 2.2.23 of Talagrand (2014)) and using the fact that for unit sphere $\mathcal{S}^{d-1}$, Talagrand's $\gamma$ functionals (see Talagrand (2014)) obey $\gamma_1(\mathcal{S}^{d-1}), \gamma_2^2(\mathcal{S}^{d-1}) \leq \mathcal{O}(d)$,

$$n^{-1}\sup_{\boldsymbol{v} \in \mathcal{S}^{d-1}} |Y_{\boldsymbol{v}}| \leq cK^2(\sqrt{d/n} + d/n + t/\sqrt{n}), \tag{F.1}$$

with probability $1 - 2\exp(-\min\{t^2, \sqrt{n}t\})$. Since $n \geq C\lambda^{-2}K^4 d$ for sufficiently large $C > 0$, picking $t = \frac{1}{16c}K^{-2}\lambda\sqrt{n}$, with probability $1 - 2\exp(-\mathcal{O}(K^{-4}\lambda^2 n))$, we ensure that right hand side of (F.1) is less than $\lambda/8$. This leads to the following inequalities

$$\frac{1}{n}\|\bar{\boldsymbol{A}}^T\bar{\boldsymbol{A}} - \mathbb{E}[\bar{\boldsymbol{A}}^T\bar{\boldsymbol{A}}]\| \leq \frac{\lambda}{8} \implies \frac{9K^2}{8}\boldsymbol{I}_d \succeq \frac{1}{n}\bar{\boldsymbol{A}}^T\bar{\boldsymbol{A}} \succeq \frac{7\lambda}{8}\boldsymbol{I}_d. \tag{F.2}$$

$$\implies \frac{9}{8}K \geq \frac{1}{\sqrt{n}}\|\bar{\boldsymbol{A}}\| \geq s_{\min}(\bar{\boldsymbol{A}}) \geq \sqrt{\frac{7}{8}\lambda}.$$

**Upper bound on spectral norm:** For spectral norm of $\boldsymbol{A}$, we use the triangle inequality

$$\frac{1}{\sqrt{n}}\|\boldsymbol{A}\| \leq \frac{1}{\sqrt{n}}(\|\boldsymbol{E}\| + \|\bar{\boldsymbol{A}}\|) \leq \max_{1 \leq i \leq n}\|\mathbb{E}[\boldsymbol{a}_i]\|_{\ell_2} + 9K/8 \leq \theta + \sqrt{3/2}K.$$

**Lower bound on minimum singular value:** This part assumes that all row expectations are same. Denote the size $n$ all ones vector by $\mathbf{1}_n$ and define the process $Z_{\boldsymbol{v}} = \frac{1}{\sqrt{n}}\mathbf{1}_n^T\bar{\boldsymbol{A}}\boldsymbol{v}$. Observe that $\bar{\boldsymbol{A}}^T\mathbf{1}_n = \sum_{i=1}^n \bar{\boldsymbol{a}}_i \in \mathbb{R}^d$ is a vector satisfying $\|\bar{\boldsymbol{A}}^T\mathbf{1}_n/\sqrt{n}\|_{\psi_2} \leq \mathcal{O}(K)$. Hence, again using $n \geq CK^4\lambda^{-2}d$ for sufficiently large $C > 0$, applying Lemma F.3 with $m = c_0 K^{-4}\lambda^2 n > d$ by picking a sufficiently small constant $c_0 > 1/C$, with probability at least $1 - 2\exp(-100c_0 K^{-4}\lambda^2 n)$

$$\frac{1}{\sqrt{n}}\sup_{\|\boldsymbol{v}\|_{\ell_2}=1}|Z_{\boldsymbol{v}}| = \frac{1}{n}\|\bar{\boldsymbol{A}}^T\mathbf{1}_n\|_{\ell_2} \leq \frac{1}{12}KK^{-2}\lambda \leq \frac{\sqrt{\lambda}}{12}.$$

Let $\boldsymbol{P} = \boldsymbol{I}_n - \frac{1}{n}\mathbf{1}_n\mathbf{1}_n^T$ be the projection onto the orthogonal complement of the all ones vector. Note that $\boldsymbol{PE}\boldsymbol{v} = 0$ as the rows of $\boldsymbol{E}$ are equal. With this observation, with desired probability, for any unit length $\boldsymbol{v}$,

$$\|\boldsymbol{A}\boldsymbol{v}\|_{\ell_2} \geq \|\boldsymbol{P}\boldsymbol{A}\boldsymbol{v}\|_{\ell_2} = \|\boldsymbol{P}\bar{\boldsymbol{A}}\boldsymbol{v}\|_{\ell_2} \geq \|\bar{\boldsymbol{A}}\boldsymbol{v}\|_{\ell_2} - |Z_{\boldsymbol{v}}| \tag{F.3}$$

$$\geq s_{\min}(\bar{\boldsymbol{A}}) - \sup_{\boldsymbol{v} \in \mathcal{S}^{d-1}}|Z_{\boldsymbol{v}}| \geq (\sqrt{7/8} - 1/12)\sqrt{\lambda n}, \tag{F.4}$$

which implies $s_{\min}(\boldsymbol{A})/\sqrt{n} \geq \sqrt{2\lambda/3}$. $\qquad\square$

The corollary below is obtained by slightly modifying the proof above by using $\frac{1}{n}\|\bar{\boldsymbol{A}}^T\bar{\boldsymbol{A}} - \mathbb{E}[\bar{\boldsymbol{A}}^T\bar{\boldsymbol{A}}]\| \leq \frac{K^2}{8}$ in line (F.2) and only focusing on the spectral norm bound.

**Corollary F.2.** *Let $\boldsymbol{A} \in \mathbb{R}^{n \times d}$ be a matrix with independent $\{\boldsymbol{a}_i\}_{i=1}^n$ subgaussian rows satisfying $\|\mathbf{zm}(\boldsymbol{a}_i)\|_{\psi_2} \leq \mathcal{O}(K)$ and $\boldsymbol{\Sigma}[\boldsymbol{a}_i] \preceq K^2\boldsymbol{I}_d$ for some $K > 0$ and $\|\mathbb{E}[\boldsymbol{a}_i]\|_{\ell_2} \leq \theta$. Suppose $\boldsymbol{\Sigma}[\boldsymbol{a}_i] \succeq \lambda\boldsymbol{I}_d$. Suppose $n \geq \mathcal{O}(K^2 d)$. Then, with probability at least $1 - 4\exp(-cK^{-2}n)$,*

$$\theta + \sqrt{3/2}K \geq \frac{1}{\sqrt{n}}\|\boldsymbol{A}\|.$$

The following lemma is fairly standard and is proved for the sake of completeness.

**Lemma F.3** (Subgaussian vector length). *Let $\boldsymbol{a} \in \mathbb{R}^n$ be a zero-mean subgaussian vector with $\|\boldsymbol{a}\|_{\psi_2} \leq L$. Then, for any $m \geq n$, there exists $C > 0$ such that*

$$\mathbb{P}(\|\boldsymbol{a}\|_{\ell_2} \leq CL\sqrt{m}) \geq 1 - 2\exp(-100m).$$

*Proof.* We can pick a $1/2$ cover $\mathcal{C}$ of the unit $\ell_2$-sphere with size $\log|\mathcal{C}| \leq 2n$. For any $\boldsymbol{v} \in \mathcal{C}$, subgaussianity implies, $\mathbb{P}(|\boldsymbol{v}^T\boldsymbol{a}| \geq t) \leq 2\exp(-\frac{ct^2}{2L^2})$. Setting $t = CL\sqrt{m}$ for sufficiently large constant $C > 0$, and union bounding over all $\boldsymbol{v} \in \mathcal{C}$, we find

$$\mathbb{P}(\bigcap_{\boldsymbol{v} \in \mathcal{C}}\|\boldsymbol{v}\|_{\ell_2} \leq CL\sqrt{m}) \geq 1 - 2\exp(2n - \frac{cC^2L^2m}{2L^2}) \leq 1 - 2\exp(-100m).$$

To conclude, let $\boldsymbol{v}(\boldsymbol{a}) \in \mathcal{C}$ be $\boldsymbol{a}$'s neighbor satisfying $\|\boldsymbol{v} - \frac{\boldsymbol{a}}{\|\boldsymbol{a}\|_{\ell_2}}\|_{\ell_2} \leq 1/2$. Hence, we have

$$\|\boldsymbol{a}\|_{\ell_2} \leq \|(\boldsymbol{a} - \boldsymbol{v}(\boldsymbol{a}))^T\boldsymbol{a}\|_{\ell_2} + \|\boldsymbol{v}^T\boldsymbol{a}\|_{\ell_2} \leq \|\boldsymbol{a}\|_{\ell_2}/2 + CL\sqrt{m} \implies \|\boldsymbol{a}\|_{\ell_2} \leq 2CL\sqrt{m}.$$

To conclude, use the change of variable $C \to C/2$. $\qquad\square$

