# OpenReview forum: "Stochastic Gradient Descent Learns State Equations with Nonlinear Activations"
_ICLR.cc/2019/Conference_

### Official Review · AnonReviewer3 · 2018-11-02
**Good convergence result for non-convex dynamic problem under stable system condition**

**Rating:** 7
**Confidence:** 3

**Review:**

The paper studies discrete time dynamical systems with a non-linear state equation.  They assume the non-linear function is assumed to be \beta-increasing like leaky ReLU. Under this setting, the authors prove that for the given state equation for stable systems with random gaussian input at each time step, running SGD on a fixed length trajectory gives logarithmic convergence.

The paper is well-written and proves strong convergence properties. The deterministic result does not seem very novel and uses the idea of one-point strong convexity which has been studied in various prior works. However the bounding of the condition number of the data matrix is interesting and guarantees are near-optimal. The faster convergence for odd activations is a good observation. Overall, I think the paper is good. I do list some concerns:
Questions/concerns:
- The deterministic theorem (Theorem 4.1) seems similar to Theorem 3 in [1] with SGD instead of GD. Also under the distribution being symmetric, it can be derived from [2] with $k=1$.
- Can the ideas be extended to other commonly used activations such as ReLUs/Sigmoids? Sigmoids have exponentially small slope near origin.
- The proof seems to rely on the fact that due to the gaussian input added each time step and stable system assumption after a sufficient number of time steps, the input-output pairs will not be highly correlated. So the data is sufficiently uncorrelated taking enough data. What happens if this data at each step is not gaussian?
- In the unstable setting, the solution proposed just samples from different trajectories which by default are independent hence correlation is not an issue, this seems a bit like cheating.
- In RNNs, the motivation of the work, the hidden vectors are not observed, thus this setting seems a bit restrictive.
- If SGD was performed on only one truncated series, do the results still hold?

Other comments:
- There has been previous work on generalized linear models which work in more general settings like GLMtron [3]. The authors should update prior work on generalized linear models as well as neural networks.
- Typo on Page 2 y_t = h_{t+1} not y_t = h_t.

[1] Dylan J. Foster, Ayush Sekhari, and Karthik Sridharan. Uniform Convergence of Gradients for Non-Convex Learning and Optimization. NIPS 2018.
[2] Surbhi Goel, Adam Klivans, and Raghu Meka. Learning One Convolutional Layer with Overlapping Patches. ICML 2018.
[3] Sham M. Kakade et al. Efficient learning of generalized linear and single index models with isotonic regression. NIPS 2011.


--------------
I would be maintaining the same score. I agree that the paper has nice convergence results that could possibly be building steps towards the harder problem of unobserved hidden states however, there is more work that could be done for unstable systems and possible extension to ReLU and other activations to take it a notch higher.

---

> ### Author Response · Authors · 2018-11-26
> **Response to Reviewer 3: Citations included and GLM discussed**
>
> Thank you for the detailed review and helpful feedback. Our response is provided below.
>
> 1 Re Similar results to Thm 4.1: We updated the manuscript and compared Thm 4.1 to [1,2]. Both results make distributional assumptions. As far as I understand, [1] is a statistical convergence result rather than algorithmic (e.g. establishing how fast SGD finds a solution). Corollary 1 of [2] is closer to Thm 4.1. The differences are [2] applies to leaky ReLU only, requires symmetric distribution and access to population landscape i.e. infinitely many samples (perhaps latter issue can be fixed using results of Mei et al. or [1]). Overall, we agree that the proof of Thm 4.1 is not that sophisticated. Actually, we asked a few colleagues to see if there is an identical result but we were unable to find it.
>
> 2 Re Extension to sigmoid and ReLU: This is a very good question (also see response to Reviewer 2). The extension to other activations has two aspects (at least using our approach): (1) Establishing SGD convergence result (similar to Thm 4.1). (2) Showing that state vector has a good covariance. There is a good chance (1) is doable using related results from literature (e.g. Soltanolkotabi 2017). The bigger challenge seems to be addressing (2) i.e. ensuring input provides enough excitation for the output state covariance. As you mention, sigmoid's slope might help (perhaps at the expense of increasing the sample size). Overall, if one assumes that the state covariance is nice (e.g. Assumption 1), we believe extension is probably doable.
>
> 3 Re Gaussian assumption: The Gaussianity assumption is again used to prove that state vector has a good covariance. The current proof needs it because it uses properties of multivariate Gaussians (e.g. rotational invariance). Again, if you assume state covariance is good (Assumption 1), one can use a wider class of distributions.
>
> 4 Re Unstable systems: We agree that creating independent trajectories simplifies the problem. The challenge is that, the past state of the unstable system is not forgotten instead it is amplified. Hence, single trajectory analysis requires a different approach. We should remark that this problem is nontrivial even without any nonlinear activation. See the results of Simchowitz et al. and Faradonbeh et al.
>
> 5 Re Truncated series: The result would still hold with truncated series or using a single subtrajectory. This is indeed the main proof strategy. We first show that truncation leads to small perturbation. Secondly, we show that if a single subtrajectory is sufficient, you can stitch them together to obtain a convergence on full dataset. As first reviewer suggested, we put a proof outline on Section 4 mentioning these.
>
> 6 Re Other GLM works: Thanks for pointing out missing refs. We added [1,2,3] and Mei et al. and compared to our results. We are happy to add other references if relevant. Some neural net works appeared after our submission. We will add these (and possible future papers) during the final revision.
>
> 7 Re Typo: Fixed. Thanks!
>
> References
> 1) Mei, Song, Yu Bai, and Andrea Montanari. "The landscape of empirical risk for nonconvex losses." The Annals of Statistics 46.6A (2018): 2747-2774.
> 2) Soltanolkotabi, Mahdi. "Learning relus via gradient descent." Advances in Neural Information Processing Systems. 2017.
> 3) Simchowitz, Max, et al. "Learning Without Mixing: Towards A Sharp Analysis of Linear System Identification." arXiv preprint arXiv:1802.08334 (2018).
> 4) Faradonbeh, Mohamad Kazem Shirani, Ambuj Tewari, and George Michailidis. "Finite time identification in unstable linear systems." Automatica 96 (2018): 342-353.

---

### Official Review · AnonReviewer2 · 2018-11-05
**Interesting and challenging problem, but assumptions weaken the results**

**Rating:** 5
**Confidence:** 5

**Review:**

This paper studies the ability of SGD to learn dynamics of a linear system + non-linear activation. That is, in the standard LTI setting, the dynamics of a system evolve according to

h_{t+1} = Ah_t + Bu_t,

on input u_t.

In addition, this paper considers the setting where the evolution is:

h_{t+1} = \phi(Ah_t + Bu_t)

for \phi a non-linear activation function.

This is a difficult problem. Though system identification was for many decades a large and active area in the control community, the understanding of system identification from a modern statistical perspective (understanding sample complexity and computational complexity simultaneously) is surprisingly lacking. This is evidenced by the fact that the first results along these lines for the simplest possible (SISO, LTI) system, came only recently (Hardt, Ma, Recht ’16).

This paper attacks a more general setting, due to the presence of the nonlinearity.

However, the present setting is significantly limited in another sense: the authors assume that the state is observed directly. This is in contrast to the typical situation where we observe only a projection of the state, or possibly even a noisy such projection. Indeed, this is one of the critical complications in the work of Hardt, Ma and Recht. Without it, i.e., under the assumption that the entire state trajectory can be directly observed, much more is possible, and indeed much more has been done. For example, work by Bento, Ibrahimi and Montanari ’10, solves a more difficult problem in that they estimate sparse dynamics (in appropriate sample complexity). Jalali and Sanghavi ’11 generalized the work of Bento et al., to the setting where some of the components of the state are not all observed, but rather some are latent.

The motivating application for this work is estimating RNNs. In this case, the state variable represents the critical information that is carried from one time to the next in the RNN. Presumably the setting here is to show that if indeed data are generated by an RNN, then we can compute this using SGD and backprop. Towards this, the assumption of having access to the internal state is a difficult one. On the one hand, this is a hard and important problem. On the other, we really won’t have access to such an internal state. There are of course other problematic aspects, such as robustness, the inability to use ReLU (Defn 3.1). But the observation model seems important. Again, I believe this is especially so, because the considerable complications present in Hardt, Ma, Rect ’16 specifically seemed to be a consequence of the observation model being partial.

The inability to use ReLU at first look does not seem like a great limitation. But then one problematic aspect here seems that the proof concept and direction critically rely on this, as they basically reduce to the setting of linear activations — something which, presumably, is impossible for something like ReLU. So it is not only the results, but also the developed machinery, that seem to be inherently limited.

This is, overall, an interesting paper, attacking an important and also very challenging area. As with all papers in this vein, we are left with having to make a judgement call on whether this simplified scenario is indeed a good first step towards solving the problems we are hoping to solve. Is it developing the right insight, right tools, etc. While I find there is a lot of interesting and good work in this paper, I am not completely convinced about this last point.

---

> ### Author Response · Authors · 2018-11-26
> **Response to Reviewer 2: Addressing assumptions**
>
> Thank you for the thorough review and bringing up good points. Our response is provided below.
>
> 1 Re State is observed directly: We agree that learning with output observations would be a more significant result. As the reviewer points out, even for linear systems, results on output observations are limited (e.g. Hardt, Ma, Recht). However, we believe that understanding nonlinear state equations is interesting in its own right. The fact that output observation is not addressed doesn't make the problem trivial. There is a very limited understanding of learning nonlinear dynamical systems and the setup considered here is a natural extension of the linear state equation. The intuitions and theory developed here can help address state equations with other nonlinearities. On the other hand, this paper also makes progress on multiple fronts which may help address output equations:
>
> (a) We show that output state has good covariance under nonlinear activations. Intuitively, this may play a critical role for learning the output observation matrix C where y_t=Cx_t.
> (b) Learning from single trajectory: We provide a technical framework for rigorously addressing temporal dependencies in the data obtained from dynamical systems.
> (c) Optimal sample complexity: We show that despite temporal dependencies and nonlinearities, one can learn dynamical systems in a statistically efficient way.
>
> 2 Re Comparison to Bento et al: We believe the results of Bento et al and Jalali and Sanghavi are orthogonal to our work rather than being more difficult. These works use convex optimization and consider sparse linear equations whereas we use first order methods for nonlinear systems. On the other hand, we believe it is possible to extend our results to sparse systems by developing sparse variants of Theorems 4.1 and F.1 of our manuscript. Finally, the sparsity dependency of these works appear to be suboptimal (as acknowledged by Jalali and Sanghavi) whereas our results are sample optimal.
>
> 3 Re Inability to use ReLU: This is a good point which is also brought up by Reviewer 3 (hence some of the response overlaps). Addressing ReLU has two challenges: (1) Establishing SGD convergence result (similar to Thm 4.1). (2) Showing that state vector has a good covariance. There is a good chance (1) is doable using related results from literature (e.g. Soltanolkotabi 2017, Mei et al. 2018 and others). The main challenge seems to be addressing (2) i.e. ensuring input provides enough excitation for the output state covariance. We believe that if one assumes that the state covariance is nice (e.g. Assumption 1), ReLU proof is likely doable. While writing the paper we had to decide between (i) obtaining a comprehensive result which provides an end-to-end learning guarantee or (ii) assuming that the state covariance is *somehow* well-conditioned and providing a more general result. We chose the first path which resulted in Gaussian input assumption as well as the necessity of beta-increasing activations.
>
> On a related note, non-rigorously speaking, leaky ReLU results may perhaps help establish a ReLU result: ReLU can be approximated by a leaky ReLU with a very small slope beta. This means that the trajectory generated by a ReLU state equation can be written as the leaky ReLU trajectory plus a small noise. Hence, we can fit the ReLU data using the leaky ReLU state equation to find the A,B matrices. It is not clear if this argument would immediately work in theory because we believe provable noise robustness of SGD (via Thm 4.1) also decays in the slope beta so you might still suffer significant error on A,B estimates (e.g. while noise is small, it gets amplified by 1/beta).
>
> References
> 1) Mei, Song, Yu Bai, and Andrea Montanari. "The landscape of empirical risk for nonconvex losses." The Annals of Statistics 46.6A (2018): 2747-2774.
> 2) Soltanolkotabi, Mahdi. "Learning relus via gradient descent." Advances in Neural Information Processing Systems. 2017.

---

### Official Review · AnonReviewer1 · 2018-11-05
**Interesting result on learning a non-linear dynamical system**

**Rating:** 7
**Confidence:** 3

**Review:**

This work considers the problem of learning a non-linear dynamical system in which the output equals the state.  Under several assumptions (input is Gaussian, non-linear activation is strictly increasing, stable system) it is shown that SGD converges linearly to the ground truth system with near-optimal sample complexity. The proof idea is to reduce this problem to the problem of learning a single non-linear neuron in the case that the covariance matrix of the data is well-conditioned. The main challenge is to show the covariance is well-conditioned under the reduction. In a nutshell, this is done by splitting the trajectory to sub-trajectories with independent states and using results from random matrix theory on matrices with independent rows.

This work tackles a very challenging problem and the results are interesting. The guarantees are strong – linear convergence to the ground truth parameters and near-optimal sample size. Given that not much is known on deep non-linear networks, I think that the result is significant. The main weakness of the paper is the assumption that the state equals the output. Another minor weakness is the clarity and presentation of results:
1.       The proof outline of the main result is hard to follow. There is no proof outline of Theorem 4.2 in the main text. The proof is highly technical and there are many technical ideas that were moved to the appendix. For instance, the proofs in sections C and D are not mentioned in the main text. I suggest to write a summary of the steps required to prove the main result and how all of the technical ideas are combined together.
2.       There is no reference and comparison to the paper of Mei et al. [1] that study single neuron models.
3.       It is claimed that by increasing beta the convergence is faster. However, I am not sure why this is meaningful. By changing beta the ground truth changes as well. For beta = 0 the ground truth dynamical system is linear and for beta = 1 the ground truth is a non-linear dynamical system with ReLU. Since a ReLU network is more expressive, generally in the case of beta = 1 the ground truth is more difficult to learn than beta = 0. Therefore, we should expect convergence to be slower than beta=0 or not occur at all. Am I missing something?
4.       The Gaussian assumption is not stated clearly in the text. It can be deduced only from the statements of the theorems and the conclusion section.
5. In Theorem F.1, it is claimed that all rows of E are equal. However, in the statement of the theorem it is not mentioned that the rows of A are identically distributed. Should this assumption be included in the statement?

[1] Mei, Song, Yu Bai, and Andrea Montanari. "The landscape of empirical risk for non-convex losses." arXiv preprint arXiv:1607.06534 (2016).‏


-----------Revision------------------------

I am not changing the score. I disagree with AnonReviewer2 regarding the significance of the results.  The assumption that the states are observed is indeed a weakness of the paper. However, understanding non-linear dynamical systems is extremely challenging and this paper provides strong convergence guarantees. Furthermore, there are several insights in the analysis that may be useful in future work.

---

> ### Author Response · Authors · 2018-11-26
> **Response to Reviewer 1: Proof outline included.**
>
> Thanks for the careful review and helpful suggestions. Really appreciated. Our response is provided below.
>
> 1 Re Proof outline: This is a really good suggestion. We updated the manuscript and added a proof outline at the start of Section 4. We believe this helps clarify the proof strategy. We also refer to Appendix A,B,C,D in this outline to navigate the reader.
>
> 2 Re Comparison to Mei et al: Thanks for catching this. I am honestly surprised that we forgot to cite this paper. The reference is included and discussed. Additionally, proof outline states that "Temporal dependencies prevent us from directly using statistical learning results that typically assume i.i.d. samples."
>
> 3 Re beta=0 vs 1 convergence: We clarified that convergence becomes faster assuming a realizable model where the data is generated from a ground truth state equation. This is explained in the Footnote 3 in the numerical section 6. In particular, our experiments are trying to verify our main theoretical results which predict that if the data is realizable then convergence will be faster with increasing beta. Hence, numerics is indeed consistent with our theory. We agree with the reviewer that, if the dataset is fixed and we try to fit this fixed dataset with different beta, the results will be different. On a related note, leaky ReLU has similar expressivity as ReLU. If a small positive slope makes it converge faster compared to ReLU, it would be interesting to know.
>
> 4 Re Gaussian not stated: We agree with the reviewer. We stated the Gaussian data assumption in the Introduction under Contributions. Technically speaking, the main use of Gaussian assumption is proving that covariance of the state matrix is well conditioned. In general, one can establish the same result for wider range of distributions (e.g. subgaussian) if Assumption 1 holds.
>
> 5 Re Theorem F.1: Thanks for carefully reading the paper. This is a really good catch. We slightly modified the statement of Theorem F.1. Basically, the spectral norm bound doesn't need the rows of E to be equal. However, the minimum singular value bound needs them to be equal. This is a minor technicality and doesn't impact the remaining proof. You can see it by reading the proof of Theorem D.2 (e.g. bounding Q and bar{Q} matrices).

---

### Meta-Review · Area_Chair1 · 2018-12-12
**ICLR 2019 decision**

**Confidence:** 4
**Recommendation:** Reject

**Metareview:**

This paper shows convergence of stochastic gradient descent  for the problem of learning weight matrices for a linear dynamical system  with non-linear activation.  Reviewers agree that the problem considered is both interesting and challenging. However the paper makes many simplifying assumptions - 1) both input and hidden state are observed, a very non standard assumption, 2) analysis requires increasing activation functions, cannot handle ReLU functions. I agree with R2 and think these assumptions make the results significantly weaker. R1 and R3 are more optimistic, but authors response does not give an insight into how one might extend this analysis to the setting where hidden state is not observed. Relaxing these assumptions will make the paper more interesting.